**Highly accurate dating of micrometre-scale baddeleyite domains through combined focused ion beam extraction and U-Pb thermal ionisation mass spectrometry (FIB-TIMS)**

Lee F White[1,2*], Kimberly T Tait[1,2], Sandra L Kamo[1,3], Desmond E Moser[4] & James R Darling[5]

[1] *Department of Natural History, Royal Ontario Museum, Toronto, Ontario, M5S 2C6, Canada*

[2] *Department of Earth Sciences, University of Toronto, Toronto, Ontario, M5S 3B1, Canada*

[3] *Jack Satterly Geochronology Laboratory, University of Toronto, Toronto, Ontario, M5S 3B1, Canada*

[4] *Department of Earth Sciences, University of Western Ontario, London, Ontario, Canada*

[5] *School of the Environment, Geography and Geosciences, University of Portsmouth, Portsmouth, PO1 3QL, UK*

*Corresponding to Lee White (lwhite@rom.on.ca)*

**Baddeleyite is a powerful chronometer of mafic magmatic and meteorite impact processes. Precise and accurate U-Pb ages can be determined from single grains by isotope dilution thermal ionisation mass spectrometry (ID-TIMS), but this requires disaggregation of the host rock for grain isolation and dissolution. As a result, the technique is rarely applied to precious samples with limited availability (such as lunar, Martian and asteroidal meteorites and returned samples) or samples containing small baddeleyite grains that cannot readily be isolated by conventional mineral separation techniques. Here, we use focused ion beam (FIB) techniques, utilising both $Xe^+$ plasma and $Ga^+$ ion sources, to liberate baddeleyite subdomains *in situ*, allowing their extraction for ID-TIMS dating. We have analysed the U-Pb isotope systematics of domains ranging between 200 μm and 10 μm in length and 5 μg to ≤ 0.1 μg in mass. In total, six domains of Phalaborwa baddeleyite extracted using a $Xe^+$-pFIB yield a weighted mean $^{207}Pb/^{206}Pb$ age of 2060.1 ± 2.5 Ma (0.12 %; all uncertainties 2σ), within uncertainty of reference values. The smallest extracted domain (ca. 10 x 15 x 10 μm) yields an internal $^{207}Pb/^{206}Pb$ age uncertainty of ± 0.37 %. Comparable control on cutting is achieved using a $Ga^+$-source FIB instrument, though the slower speed of cutting limits potential application to larger grains. While the U-Pb data are between 0.5 and 13.6 % discordant, the extent of discordance does not correlate with the ratio of material to ion-milled surface area and results generate an accurate upper intercept age in U-Pb concordia space of 2060.20 ± 0.91 Ma (0.044 %). Thus, we confirm the natural U-Pb variation and discordance within the Phalaborwa baddeleyite population observed with other geochronological techniques. Our results demonstrate the FIB-TIMS technique to be a powerful tool for highly accurate *in-situ* $^{207}Pb/^{206}Pb$ (and potentially U-Pb in concordant materials) age analysis,**

**allowing dating of a wide variety of targets and processes newly accessible to geochronology.**

*Keywords. FIB-TIMS; FIB; TIMS; U-Pb; Baddeleyite; Geochronology; In-situ*

## 1.0 Introduction

The generation of high precision chronological data is a cornerstone of the Earth and planetary sciences, providing an absolute measurement on which to anchor relative observations of geological time (e.g. Gradstein et al., 2004). The most precise radiogenic isotopic ratios (e.g. U-Th-Pb, Sm-Nd, Rb-Sr) are generated using isotope dilution thermal ionisation mass spectrometry (ID-TIMS; Parrish and Noble, 2003), which has been used to measure the timing of Solar System formation (Amelin et al., 2002), initial differentiation of the Moon (Barboni et al., 2017), and the timing of crustal formation on Mars (Bouvier et al., 2018), often with internal age uncertainties on the order of ~0.1% 2σ. In particular, U-Pb isotopic measurements of the accessory minerals zircon ($ZrSiO_4$) and baddeleyite ($ZrO_2$) by ID-TIMS allows for direct, high precision dating of magmatic, metamorphic, and shock metamorphic events (e.g. Krogh et al., 1987; Parrish and Noble., 2002; Bouvier et al., 2018).

To attain the high level of precision and accuracy offered by ID-TIMS, desired mineral grains must be isolated from their host rock using crushing and sieving techniques or electric pulse disaggregation before separation based on density, magnetic and optical properties of the target mineral phase (e.g. Söderlund and Johansson, 2002). As a result, the analysed grains preserve no evidence of their petrological or mineralogical context and are challenging to characterise prior to dating, which makes the accurate interpretation of U-Th-Pb ages in samples with complex thermal, metamorphic and deformational histories highly challenging (e.g. Krogh et al., 1993a; Krogh et al., 1993b; Parrish and Noble., 2002; Bouvier et al., 2018). In addition, the small grain size (commonly < 50 μm) and bladed nature of individual baddeleyite crystals makes clean separation of target grains time consuming and highly challenging (e.g. Söderlund and Johansson, 2002). Though grains can be chemically or physically abraded to remove potentially discordant crystallographic domains (Krogh, 1982; Rioux et al., 2010) or physically broken to yield isolated fragments (Amelin, 1998), ID-TIMS is incapable of separating crystallographic domains of potentially different ages, such as micrometre-scale recrystallized or altered domains in shocked minerals which may record disturbed U-Pb isotope reservoirs (Cavosie et al., 2015; White et al., 2017a,b). These realities mean that, although ID-TIMS is

the method that can deliver the most accurate and precise isotopic data, it has historically
remained challenging to impossible to apply it to rare meteoritic or returned planetary
materials, or mineral targets located within thin sections. Developing the capability for highly
targeted extraction of micrometre-scale domains for ID-TIMS would be a powerful
advancement. It would permit the generation of relatively highly precise radio-isotopic ages
from microstructurally and chemically characterised grains (Moser et al., 2011, 2013; Darling
et al., 2016). In effect, *in situ* dating using petrological and mineralogical evidence of
crystallization and/or metamorphic history, information that is often critical to accurately
interpreting isotopic ages.

Focused ion beam (FIB) technologies are a staple of the material sciences, most commonly
used to fabricate and analyse nanomaterials (Matsui et al., 2000; Schaffer et al., 2012). Within
the Earth and planetary sciences, FIB's have principally been used to prepare thin foils for
analysis of materials by transmission electron microscopy (TEM), which requires a sample to
be electron transparent (Heaney et al., 2001), and the preparation of microtip specimen for
atom probe tomography (e.g. Reddy et al., 2016). Although $Ga^+$ source FIB's are the most
common, the linear relationship between beam current and spot size prevents operation of the
instrument at high currents (> 20 nA), limiting the rate (and thus volume) of material removal
to the tens of micrometres in a single day session. Options for the removal of larger masses
require higher energy; for example, laser cutting allows extraction of millimetre-scale sections
of material but induces deeper and more severe damage to the milled surface (Echlin et al.,
2012), and in the case of geological materials may result in localised fractionation of target
elements and isotopes comparable to heating effects seen in laser ablation inductively coupled
plasma mass spectrometry (LA-ICP-MS; e.g. Košler et al., 2005; Ibanez-Mejia et al., 2014).
Such side effects are not induced by micro-drill extraction of target phases (e.g. Paquette et al.,
2004), but the spatial resolution offered by such an approach is incapable of isolating
exceptionally small (< 50 μm) domains, such as meteoritic micro-baddeleyite (Herd et al.,
95  2018).


Recent advances in FIB technologies have significantly broadened the range of ion beam
chemistries and source types, the most recent being the magnetically enhanced, inductively
coupled xenon ($Xe^+$) plasma ion source (Bassim et al., 2014). While a $Ga^+$ liquid metal ion
source (LMIS) FIB loses spatial resolution at higher currents (I) due to spherical aberration
resulting in a non-Gaussian beam shape with large tails (Smith et al., 2006; Bassim et al., 2014),
the $Xe^+$ ICP source remains stable beyond ~60 nA. As a result, a finer spot size can be achieved
for the same focussing optics using a $Xe^+$ pFIB, while the superior angular intensity allows for
high current milling as the effects of spherical aberration are not realised (Smith et al., 2006).
This makes the attainment of currents in the µA range possible with a $Xe^+$ pFIB, which cannot
be achieved with a $Ga^+$ LMIS instrument whilst retaining a focused beam (Figure 1). Another
important benefit of the $Xe^+$ pFIB is a direct result of the larger ionic size of $Xe^+$ compared to
$Ga^+$ (e.g. Yuan et al., 2017), which results in more atoms of material being ejected from the
target per incident ion and yielding a higher removal rate. Though sputtering rates are typically
on the order of 10 - 30% higher for $Xe^+$ compared to $Ga^+$, Cu (~300% higher) and Si (30 - 50%
higher) demonstrate notably higher sputter rates per coulomb when exposed to a $Xe^+$ ion beam
(Ziegler et al., 1985). The larger ionic size of $Xe^+$ also results in a shallower depth of ion
penetration and resulting damage to the target material; for example, the penetration of $Ga^+$ and
$Xe^+$ in Si is 24 nm and 28 nm respectively at 30 kV (Ziegler et al., 1985; Burnett et al., 2016).
However, the effect of this interaction, which often results in amorphisation of the surface layer
exposed to the ion beam, on trace element distribution and mobility is poorly constrained. For
example, the effect on U and Pb mobility is unknown. In this study, we analyse multiple
samples of the Phalaborwa U-Pb baddeleyite reference material, which have been extracted *in-*
*situ* via $Ga^+$ FIB, $Xe^+$ pFIB, and by mechanical (non-FIB) fragmentation to test for structural
damage, heating and ion implantation during FIB exposure, establishing a new approach to
micro-sampling for high precision ID-TIMS analysis and demonstrating the potential for $Xe^+$
FIB techniques to extract grains from thin section.

**2.0 Sample and methodology**

Originally sampled from the Phalaborwa complex (a composite intrusion of cumulate
clinopyroxenites related to pulsed carbonatite magma emplacement) in South Africa,
baddeleyite grains from the locality are often used as a reference material in U-Th-Pb studies
(Reischmann, 1995; Heaman, 2009; Schmitt et al., 2010). A single large crystal of Phalaborwa
baddeleyite was acquired from the same sample at the Royal Ontario Museum as studied by
(Heaman, 2009), who undertook 68 ID-TIMS measurements of 2 to 384 mg fragments of this
material. These fragments are variable in U concentration (51 - 2124 ppm) and the majority of
U-Pb analyses from Heaman (2009) are <1 % discordant, although individual analyses are up
to 10% discordant. A precise weighted mean $^{207}Pb/^{206}Pb$ age from 56 baddeleyite analyses of
2059.70 ± 0.35 Ma from that study, though with significant scatter (MSWD = 12), is taken as
the best measure of the crystallization age. Variations in U content and U-Pb age have also
been reported during high spatial resolution isotopic analyses of Phalaborwa, such as depth
profiling laser ablation inductively coupled plasma mass spectrometry (LA-ICP-MS) (Ibanez-
Mejia et al., 2014). During 326 small volume LA-ICP-MS analyses of Phalaborwa baddeleyite,
U concentration (87 - 1478 ppm) and percentage discordance (<13.7 %) vary substantially,
while the majority (77%) of U-Pb analyses are >1 % discordant outside of uncertainty (Ibanez-
Mejia et al., 2014). Notably, 30 of these analyses are highly discordant (>5 % discordance).

### 2.1 Focused ion beam (FIB) extraction of target domains

A large (~5 cm) grain of Phalaborwa baddeleyite was taken from the mineralogy collection at
the Royal Ontario Museum, Toronto, Canada, for use in this study (accession number
M37144). The grain was mounted in epoxy and polished to expose the surface of the grain
using 6 μm, 1 μm and 0.5 μm grit diamond paste. The epoxy mount was secured to an SEM
stub and coated with a 15 nm thick carbon coat prior to imaging and FIB work. A Thermo
Scientific Helios G4 UXe DualBeam pFIB at the Canadian Centre for Electron Microscopy
(CCEM) in McMaster University, Canada, and a Hitachi NB5000 Ga-FIB at the Ontario Centre
for the Characterisation of Advanced Materials (OCCAM) in the University of Toronto,
Canada, were used in this study.

The Xe-pFIB was operated at 30 kV, 2.5 μA for the largest cuts, facilitating the extraction of a
100 x 100 x 100 μm cube domain of baddeleyite in 32 minutes and two 200 x 50 x 30 μm
rectangular domains in 21 minutes each. Two small (5 x 15 x 10 μm) cuboids of baddeleyite
were completely isolated by 2 minutes of Xe-pFIB exposure. The Ga-FIB was operated at 40
kV and >50 nA (estimated current from previous calibration), taking two hours to isolate a 50
x 50 x 50 μm cube domain. In all scenarios a small amount of material (< 5 μm wide) was left
to anchor the isolated domain to the host mount (Figure 2). This allowed transportation of the
grain mount to the Jack Satterly Geochronology Laboratory for extraction without the use of
platinum or tungsten weld, which may contain unconstrained levels of common Pb. The grain
mount was placed in a large petri dish before being entirely submerged in ethanol. Fine tipped
tweezers and custom pipettes (made within the Jack Satterly Geochronology Laboratory) were
used to physically detach the FIB'ed material under an optical microscope, where it became
suspended in the alcohol layer before being transferred to a separate dish for imaging.
Following extraction, the tip of one of the 200 x 50 x 30 μm rectangles was physically broken
to produce two smaller (< 15 μm) domains consisting of outer surface areas that have been
both FIB'ed and not FIB'ed. Extraction of FIB'ed domains was augmented by gouging six
chips (200 μm to 3 mm in size) of material from the same mounted grain to test the larger-scale
homogeneity of the target material, which were separated into two aliquots, and two smaller
grains separated and supplied as an existing U-Th-Pb reference material were also analysed. In
total, twelve TIMS analyses are incorporated into this study; two whole (< 40 μm) baddeleyite
grains, and ten subsampled domains of the large mounted grain; two aliquots of material
physically carved out with no FIB exposure, one subdomain with Ga-FIB extraction, and seven
using Xe-pFIB extraction.

*2.2 U-Pb thermal ionisation mass spectrometry (TIMS)*
U–Pb geochronology was conducted at the Jack Satterly Geochronology Laboratory,
Department of Earth Sciences, at the University of Toronto. Grain weights were estimated from
photomicrographs, aided by known size dimensions of grains from FIB preparation. The grains
were cleaned in room temperature 8N $HNO_3$ on parafilm using a micropipette before being
loaded into dissolution vessels with a mixed $^{205}Pb$ – $^{235}U$ isotopic tracer solution. Baddeleyite
was dissolved using ~0.10 ml of concentrated hydrofluoric acid (HF) and ~0.02 ml of 8N nitric
acid ($HNO_3$) at 200° C (Krogh, 1973) for up to 5 days, then dried to a precipitate, and re-
dissolved in ~0.15 ml of 3N hydrochloric acid (HCl). Uranium and lead were isolated from the
solutions using anion exchange chromatography, dried in dilute phosphoric acid ($H_3PO_4$), and
deposited onto outgassed rhenium filaments with silica gel (Gerstenberger and Haase, 1997).
U and Pb were analysed with a VG M354 mass spectrometer in dynamic mode with a Daly
pulse-counting system. The dead time of the Daly measuring system for Pb and U was 16.5
and 14.5 ns, respectively, determined using standard reference materials 982 and U500,
respectively. The mass discrimination correction for the Daly detector is constant at 0.05
%/atomic mass unit. Thermal mass fractionation was corrected using 0.1% per atomic mass
unit for both Pb and U. Given the apparent pristine nature of the FIB-extracted baddeleyite
domains, the total common Pb in each baddeleyite analyses was attributed to laboratory Pb
(corrected using an isotopic composition of $^{206}Pb/^{204}Pb$ of 18.49 ± 4.0 %, $^{207}Pb/^{204}Pb$ of 15.59
± 4.0%, $^{208}Pb/^{204}Pb$ of 39.36 ± 4.0 %; 2σ uncertainties), thus no correction for initial common
Pb from geological sources was made. Routine testing indicates that laboratory blanks for Pb
and U are   usually less than 0.5 and 0.01 pg, respectively. Corrections to the $^{206}Pb/^{238}U$ and
$^{207}Pb/^{206}Pb$ ages for initial $^{230}Th$ disequilibrium have been made assuming a Th/U ratio in the
magma of 4.2, based on assumed crustal average values. Decay constants are those of (Jaffey
et al., 1971) ($^{238}U$ and $^{235}U$ are 1.55125 x $10^{-10}$ and 9.8485 x $10^{-10}$ per year, respectively).  A U
isotopic composition of 137.818 was used (Hiess et al., 2012). All age errors quoted in the text
and tables, and error ellipses in the concordia diagram are given at 2σ.

**3.0 Results**
In total, eleven TIMS analyses were conducted on grains and subdomains of the Phalaborwa
baddeleyite, with one sample extracted by $Ga^+$ FIB, six using the $Xe^+$ pFIB, two with no
exposure to the FIB instruments and two entirely separate whole grains. A summary of the U-
Pb isotopic data is presented in Table 1. U concentrations vary widely between 106 and 3027
ppm, in agreement with published values for Phalaborwa baddeleyite (Heaman, 2009; Ibanez-
Mejia et al., 2014; Reinhard et al., 2018),  indicative of highly variable U concentrations within
individual grains of the Phalaborwa baddeleyite (Ibanez-Mejia et al., 2014; Reinhard et al.,

214    2018).


The total amount of common Pb in several of our U-Pb analyses exceeded the estimated Pb
procedural blank of 0.5 picograms. All initial common Pb in our analytical data was corrected
using the reported laboratory blank isotopic composition under the assumption that common
Pb was introduced during laboratory procedures. Although no micro-inclusions or fracture-
hosted alteration zones (that may contain Pb) were observed using an SEM or optical
microscope, it is possible that measured common Pb is geological in origin. To test the impact
of our common Pb selection we conducted sensitivity tests for three of our results that contain
the highest amounts of total common Pb (analyses 4, 5, 10, Table 1). These show that by
applying model-based corrections to the initial common Pb above our assumed procedural
blank, has a negligible effect on the $^{207}Pb/^{206}Pb$ weighted mean age of the 11 analyses. Using
a crustal Pb isotopic composition (Stacey and Kramers, 1975) and mantle Pb isotopic
composition ($^{206}Pb/^{204}Pb$ of 14.624, $^{207}Pb/^{204}Pb$ of 15.038) produced age offsets of 0.0044%
and 0.0019%, respectively (i.e., from a mean of 2060.14 ± 0.88 Ma (n=11) to 2060.05 ± 0.81
Ma and 2060.18 ± 0.89 Ma, respectively).

Two separate ~30 μm crystals, independent of the large grain embedded for FIB work, yield
near concordant results with uncertainties on the order of ± 0.07% 2σ (analyses 1-2, Table 1).
Two larger (~100 μm) domains physically broken out of the baddeleyite mounted in epoxy
give similar $^{207}$Pb/$^{206}$Pb ages of 2060.9 Ma and 2061.6 Ma Ma (± 0.09%), though display < 8.4
% discordance in the measured U-Pb systematics (with a youngest $^{206}$Pb/$^{238}$U age of 1912 Ma;
analyses 3-4; Table 1).

All domains extracted by FIB (both Ga$^+$ and Xe$^+$ source) yield high precision $^{207}$Pb /$^{206}$Pb ages
(0.07% - 0.4 %) that are in agreement with published TIMS and SS-LA-ICPMS values
(Heaman, 2009; Ibanez-Mejia et al., 2014). This includes a small flake (analysis 7; Table 1)
containing as little as ~4.5 pg of Pb. However, all data points, aside from the two whole grains
and 5 chips from the mount (analyses 1-3; Table 1), are discordant (Figure 3). The most
discordant analysis (13.6% discordant, $^{206}$Pb/$^{238}$U age of 1811 Ma) was generated by the
smallest (5x15 μm) domain isolated by Xe-pFIB (analysis 11, Table 1), though there is
otherwise no correlation between surface area exposed to the FIB and severity of discordance.
This is supported by the observed age overlap between the 50 μm$^3$ cube prepared by Ga$^+$ FIB
(analysis 5) and 100 μm$^3$ cube prepared by Xe$^+$ pFIB (analysis 9), which yield U-Pb ages with
5.1 and 6.2% discordance, respectively. Plotting all data, with the exception of the most
discordant datum, on a concordia diagram (n = 10) produces a discordant array with intercepts
of 2060.20 ± 0.91 Ma and -5 ± 36 Ma (MSWD = 0.99) while all Ga$^+$ and Xe$^+$ FIB-TIMS data
points (*n* = 7) produce intercepts of 2062.8 ± 5.8 Ma and 80 ± 150 Ma (MSWD = 3.6) and a
weighted mean $^{207}$Pb/$^{206}$Pb age of 2060.0 ± 2.1 Ma (MSWD = 4).

**4.0 Discussion**
*4.1 FIB extraction for U-Pb isotopic analysis*
The effects of both the Ga$^+$ and Xe$^+$ FIB instruments on the U-Pb isotope systematics in
accessory phase geochronometers has never been explored, and thus the potential for the ion
beam to induce Pb diffusion and loss in exposed surface areas must be addressed for the FIB-
TIMS technique. Previous studies utilising the extraction of baddeleyite domains using FIB
instruments (such as for structural and isotopic analysis by atom probe tomography (APT);
Reinhard et al., 2018; White et al., 2017a,b) provide a poor comparison, given the application
of a low-energy (~40 pA, 5 kV) final polish to remove material that may have been damaged
or implanted with Ga-ions during interaction with the beam. While our new FIB-TIMS data
are up to 13.6 % discordant, there is no obvious correlation between the severity of discordance
and the method used to isolate the domain for TIMS dating. For example, domains physically
broken away from the mount (e.g. with no exposure to either the $Xe^+$ or $Ga^+$ FIB beam) have
$^{206}Pb/^{238}U$ ages of 2052 ± 4 Ma and 1912 ± 3.4 Ma (analyses 3-4, Table 1), representing the
oldest and second youngest measured age of the large Phalaborwa crystal incorporated into this
study. The lack of correlation between measured discordance, FIB ion source ($Ga^+$ or $Xe^+$),
FIB exposure time, and subsampled domain size (Figure 4) provides strong evidence that the
extracted domains represent natural heterogeneity of the large Phalaborwa crystal, and not
localised FIB-induced mobilisation and loss of Pb or other effects related to implantation of
the primary ion beam. This observation supports previous studies into FIB induced damage in
materials, which despite inducing up to 22 nm of surface amorphisation has never been reported
to induce local isotopic or elemental fractionation in the target material (Schaffer et al., 2012;
Burnett et al., 2016). Furthermore, the FIB-TIMS method had not led to significantly higher
procedural Pb blanks compared to standard chemistry, further supporting the ability of FIB
instruments to produce TIMS samples free of contamination and localised elemental
fractionation.

*4.2 Isotopic heterogeneity in Phalaborwa baddeleyite*
Single shot laser ablation inductively coupled plasma mass spectrometry (SS-LA-ICP-MS)
work on Phalaborwa has revealed discrepancy from measured TIMS Pb/Pb ages of between
0.1 and 2.6%, and discordance in U-Pb systematics of up to 13.7 % (Ibanez-Mejia et al., 2014).
Sub micrometre scale variations in the $^{206}Pb/^{238}U$ ratio have also been reported by atom probe
analyses of Phalaborwa baddeleyite (White et al., 2017b; Reinhard et al., 2018). $^{206}Pb/^{238}U$ ages
generated by SS-LA-ICP-MS (Ibanez-Mejia et al., 2014) hint at variations in age of the
Phalaborwa baddeleyite reference material, though the low precision of these data points (<
8.6%) may partially mask local heterogeneities. By subsampling a single large grain of
Phalaborwa baddeleyite, we observe that measured U-Pb ages vary by up to 227 Ma, and Pb/Pb
ages vary by less than 6 Ma. It is likely that the small volumes analysed by FIB-TIMS (and
secondary ionisation mass spectrometry (SIMS); c. 10 x 10 x 1 μm) act to subsample natural
U zonation and variation within the Phalaborwa baddeleyite standard that are otherwise
homogenised during larger volume analyses (e.g. whole grain TIMS or LA-ICP-MS). Care
must be taken to select pristine subdomains of material when using the Phalaborwa baddeleyite
as a small-volume U-Pb mineral standard, particularly for techniques such as FIB-TIMS or
atom probe tomography (Reinhard et al., 2018).

An additional possible source of discordance in baddeleyite U-Pb TIMS analysis is the
incorporation of zircon overgrowths (which are subjected to Pb loss; Davidson and van
Breeman, 1998; Rioux et al, 2010; Pietrzak-Renaud and Davis, 2014) or surrounding common-
Pb bearing phases in the extracted volume. This is not an issue in FIB-TIMS as such features
can be removed using the FIB instrument prior to U-Pb analysis (Figure 5). While such work
will significantly improve the concordance of generated U-Pb ages, it will also reduce the
volume of material than can be analysed by TIMS, potentially increasing the risk of grain loss
during extraction and manipulation.

*4.3 Minimum sample sizes accessible by FIB-TIMS*
With the development of the $Xe^+$ pFIB, the FIB-TIMS technique can be applied to *in-situ* target
mineral grains up to millimetres in size (e.g. Burnett et al., 2016). It is also possible to isolate
domains as small as ~5 μm, though manipulating such small regions under optical microscope
(e.g. for acid dissolution prior to ID-TIMS) is challenging and can result in the loss of extracted
grains. At the smallest grain sizes, ejection of daughter Pb atoms from crystal surfaces through
direct alpha recoil ejection can result in discordant U-Pb ages from the outermost 24 nm (± 7
nm) of the baddeleyite crystal (Davis and Davis, 2017). This would only become an issue when
sampling small grains (<15 μm thick) in their entirety, as the large surface-area to volume ratio
would potentially lead to slightly discordant U-Pb ages following extensive ejection of
daughter isotopes (Romer, 2003), requiring a simple linear correction on the order of 0.1 - 0.5%
(Davis and Davis, 2017). Subsampling internal domains of larger grains will circumvent this
issue, allowing the targeted extraction of centralised regions which are unlikely to have ejected
Pb during an alpha recoil event.

At the smallest sample sizes, uncertainties will naturally start to increase due to the reduced
atoms / counts of U and Pb. However, we demonstrate that even in the smallest baddeleyite
domains analysed here (0.05 μg; ~10 μm length) uncertainties on the corrected $^{206}Pb/^{238}U$ ages
do not rise above ± 0.85%. Associated $^{207}Pb/^{206}Pb$ ages display ± 0.38 % 2σ uncertainties. The
FIB-TIMS technique also acts to circumvent any variability in measured U-Pb ratios (< 5%)
induced by orientation-dependent Pb/U fractionation during secondary-ion mass spectrometry
(Wingate and Compston, 2000; Schmitt et al., 2010) as the high energy of the FIB instrument
(~2.5 μA) would not induce preferential channelling of ions along low-index crystal lattice
orientations, comparable to laser ablation inductively coupled plasma mass spectrometry
analysis (Ibanez-Mejia et al., 2014).

**5.0 Conclusions**

We have shown that volumes as small as ~5 x 15 μm can be effectively isolated, extracted and
dated *in-situ* using the FIB-TIMS technique developed for this study. From these tiny domains,
an accurate upper intercept U-Pb age ($2060.2 \pm 0.91$ Ma, $2\sigma$) and weighted average Pb/Pb age
($2060.29 \pm 0.57$ Ma $2\sigma$) can be generated. Both $Ga^+$ and $Xe^+$ source focused ion beams were
employed, and while we find no evidence of isotopic fractionation within the target material
using either instrument we recommend using a $Xe^+$ pFIB where possible due to the order-of-
magnitude faster mill rates, particularly if applying this technique to larger (> 50 μm) mineral
grains and subdomains. Using the FIB-TIMS technique, it is now possible to produce high
precision ages from mineral grains that have been extensively imaged and characterised within
a thin section, though extra care must be taken during the physical extraction of the smallest
domains. This technique will be of particular importance for meteoritic and returned samples,
which are too valuable to be exposed to the destructive protocol typically required for TIMS
analysis and will allow the generation of high precision age data from accessory phases
previously inaccessible to geochronology.

**Author Contributions**

L.F.W. and J.R.D conceived the study. L.F.W and S.L.K. directed and conducted the
experiments. K.T.T and D.E.M provided materials. All authors interpreted the data. L.F.W
drafted the manuscript with input from all co-authors.

**Acknowledgments**

L.F.W is supported by a Hatch Ltd. Postdoctoral fellowship. D.E.M., S.L.K., and K.T.T. are
supported by NSERC Discovery Grants. This study was supported by an STFC grant to JRD
(ST/S000291/1). We thank Ian Nicklin, Tanya Kizovski, Ana Černok (ROM), Brian Langelier
(McMaster University) and Gabriel Arcuri (Western University) for useful discussions on
technique development and implementation. We thank Sal Boccia, Jane Howe (University of
Toronto) and Travis Casagrande (McMaster University) for assistance with and access to the
Ga$^+$ and Xe$^+$ focused ion beam instruments incorporated into the study.

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

| No. | Fraction | Weight (µg) | U (ppm) | Th/U [a] | Pb_c (pg) [b] | $^{206}$Pb/$^{204}$Pb measured [c] | $^{207}$Pb/$^{235}$U [d] | 2σ | $^{206}$Pb/$^{238}$U [d] | 2σ | error corr [e] | $^{207}$Pb/$^{206}$Pb [d] | 2σ | $^{206}$Pb/$^{238}$U [e] | 2σ | $^{207}$Pb/$^{235}$U [e] | 2σ | $^{207}$Pb/$^{206}$Pb [e] | 2σ | % Disc [f] |
|---|---|---|---|---|---|---|---|---|---|---|---|---|---|---|---|---|---|---|---|---|
| 1 | Whole grain #1 | 0.60 | 431 | 0.02 | 0.17 | 36226 | 6.580 | 0.016 | 0.37480 | 0.00076 | 0.944 | 0.127333 | 0.000106 | 2052.0 | 3.6 | 2056.7 | 2.1 | 2061.5 | 1.5 | 0.5 |
| 2 | Whole grain #2 | 1.50 | 277 | 0.01 | 0.18 | 54821 | 6.574 | 0.015 | 0.37458 | 0.00071 | 0.940 | 0.127281 | 0.000105 | 2050.9 | 3.3 | 2055.8 | 2.0 | 2060.8 | 1.5 | 0.6 |
| 3 | 5 chips from mount | 0.80 | 1591 | 0.01 | 1.09 | 28030 | 6.581 | 0.017 | 0.37479 | 0.00086 | 0.908 | 0.127345 | 0.000137 | 2051.9 | 4.0 | 2056.8 | 2.3 | 2061.6 | 1.9 | 0.6 |
| 4 | 1 chip from mount | 1.30 | 3027 | 0.01 | 12.3 | 7104 | 6.066 | 0.015 | 0.34531 | 0.00072 | 0.920 | 0.127400 | 0.000132 | 1912.2 | 3.4 | 1985.3 | 2.2 | 2062.4 | 1.8 | 8.4 |
| 5 | 50x50x50 um cube (Ga⁺-FIB) | 0.40 | 443 | 0.02 | 1.91 | 2143 | 6.261 | 0.023 | 0.35705 | 0.00092 | 0.779 | 0.127179 | 0.000299 | 1968.2 | 4.4 | 2013.0 | 3.3 | 2059.3 | 4.2 | 5.1 |
| 6 | Flake (subset of rectangle #2) (Xe⁺-p FIB) | 0.20 | 106 | 0.01 | 0.33 | 1163 | 6.487 | 0.055 | 0.37174 | 0.00267 | 0.885 | 0.126561 | 0.000503 | 2037.6 | 12.5 | 2044.1 | 7.5 | 2050.7 | 7.0 | 0.7 |
| 7 | Flake (subset of rectangle #2) (Xe⁺-p FIB) | 0.05 | 254 | 0.01 | 0.67 | 464 | 6.413 | 0.066 | 0.36649 | 0.00356 | 0.906 | 0.126902 | 0.000557 | 2012.8 | 16.8 | 2034.0 | 9.1 | 2055.5 | 7.7 | 2.4 |
| 8 | 200x50x50 um rectangle #1 (Xe⁺-p FIB) | 2.50 | 284 | 0.02 | 0.66 | 24743 | 6.248 | 0.017 | 0.35613 | 0.00085 | 0.937 | 0.127252 | 0.000120 | 1963.8 | 4.0 | 2011.3 | 2.3 | 2060.4 | 1.7 | 5.4 |
| 9 | 100x100x100 um cube (Xe⁺-p FIB) | 5.00 | 397 | na | 0.54 | 83466 | 6.201 | 0.015 | 0.35327 | 0.00073 | 0.954 | 0.127302 | 0.000100 | 1950.2 | 3.5 | 2004.5 | 2.1 | 2061.1 | 1.4 | 6.2 |
| 10 | 200x50x50 um rectangle #2 (Xe⁺-p FIB) | 2.50 | 352 | 0.02 | 1.28 | 15456 | 6.161 | 0.015 | 0.35104 | 0.00073 | 0.908 | 0.127288 | 0.000129 | 1939.6 | 3.5 | 1998.9 | 2.1 | 2060.9 | 1.8 | 6.8 |
| 11 | 5x15um domain (Xe⁺-p FIB) | 0.08 | 510 | 0.04 | 0.25 | 3499 | 5.672 | 0.023 | 0.32430 | 0.00115 | 0.903 | 0.126843 | 0.000218 | 1810.7 | 5.6 | 1927.1 | 3.4 | 2054.7 | 3.0 | 13.6 |

NOTES:

(a) Th/U calculated from radiogenic $^{208}$Pb/$^{206}$Pb ratio and $^{207}$Pb/$^{206}$Pb age assuming concordance.

(b) Pb_c is total common Pb assuming the isotopic composition of laboratory blank ($^{206}$Pb/$^{204}$Pb=18.49±4%; $^{207}$Pb/$^{204}$Pb=15.59±4%; $^{208}$Pb/$^{204}$Pb=39.36±4%).

(c) $^{206}$Pb/$^{204}$Pb corrected for fractionation and common Pb in the spike.

(d) Pb/U and $^{207}$Pb/$^{206}$Pb ratios corrected for fractionation, common Pb in the spike, and blank.

(e) correction for $^{230}$Th disequilibrium in $^{206}$Pb/$^{238}$U and $^{207}$Pb/$^{206}$Pb assuming Th/U of 4.2 in the magma.

(f) error corr is correlation coefficients of X-Y errors on the concordia plot.

(f) disc is percent discordance for the given $^{207}$Pb/$^{206}$Pb age.

Decay constants are those of Jaffey et al. (1971): $^{238}$U and $^{235}$U are 1.55125 X 10$^{-10}$/yr and 9.8485 X 10$^{-10}$/yr.

$^{238}$U/$^{235}$U ratio of 137.818 was used (Hiess et al., 2012).

***Table 1:*** *U-Pb isotopic data for FIB-extracted baddeleyite and whole grains and fragments from the Phalaborwa carbonatite.*

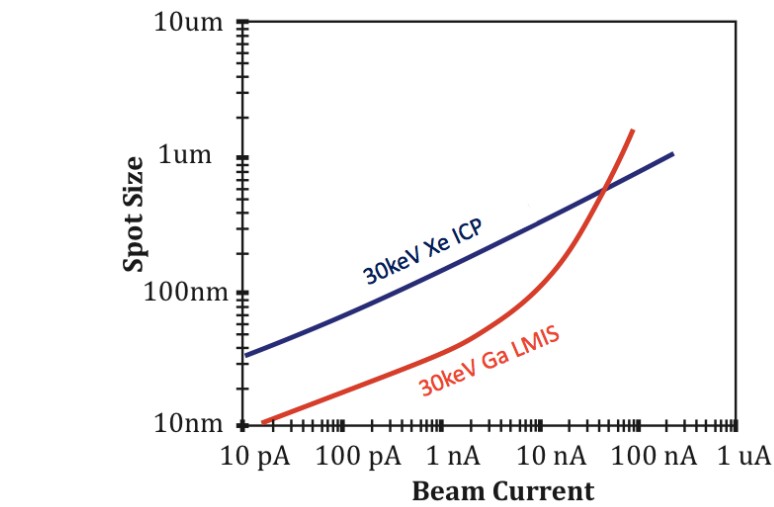

**541**
**542**
**543**
**544**
**545**
**546**
**547**
**548**
**549**
**550**
**551** ***Figure 1:*** *Spot size versus beam current for Xe⁺ and Ga⁺ source focused ion beam (FIB)*
**552** *instruments. At higher beam currents (> 10 nA) the spot size generated by the liquid metal*
**553** *ion source (LMIS) Ga⁺ FIB exponentially increases due to spherical aberration, limiting the*
**554** *energy that can be applied during milling. The inductively coupled Xe pFIB source remains*
**555** *stable at higher currents, yielding a linear increase in spot size with beam current and*
**556** *allowing higher energies to be applied without sacrificing spatial precision. This opens the*
**557** *door to larger scale (millimetre) milling experiments, such as extracting whole mineral*
**558** *phases from thin section or grain mount. Adapted after (Burnett et al. 2016).*
**559**
**560**
**561**
**562**
**563**
**564**
**565**
**566**
**567**
**568**
**569**
**570**
**571**

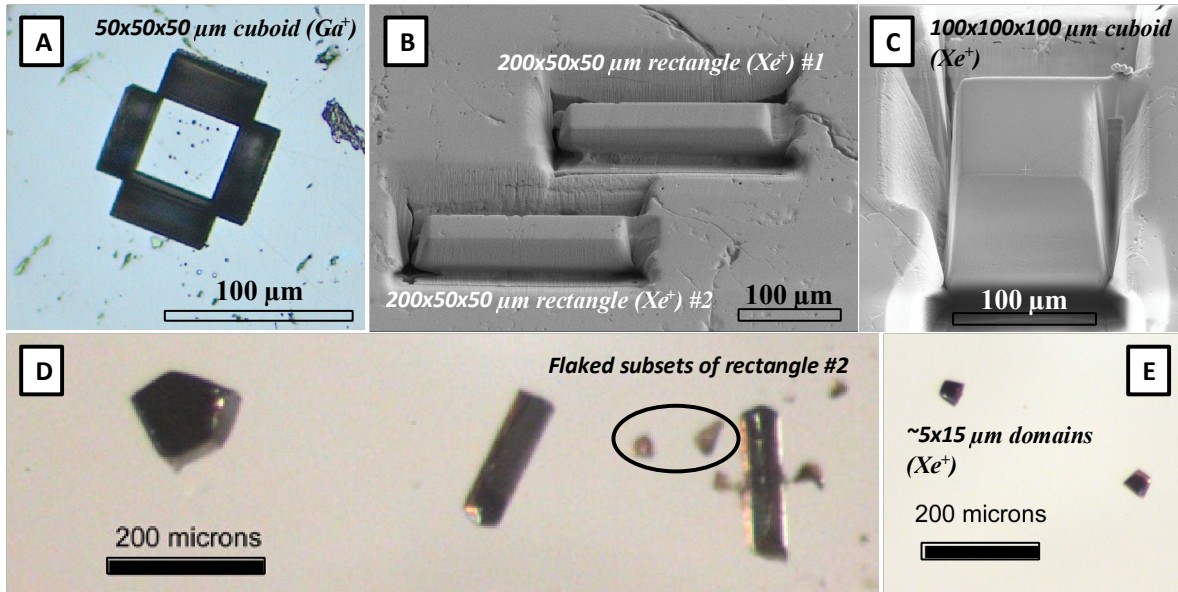


*Figure 2:* Optical microscopy and secondary electron (SE) imaging of isolated baddeleyite domains in Phalaborwa baddeleyite mount. The small amount of material left to anchor the domains (*A - C*) is critical in transporting the mount without losing material, and in ensuring easy extraction without the need for tungsten weld or complicated and time-consuming micro-manipulator usage. Once released from the grain mount, samples can be broken into further sub samples (*D*) and extensively images (*E*). All of the FIB extracted samples used for TIMS analyses, as denoted in table 1, are imaged here.


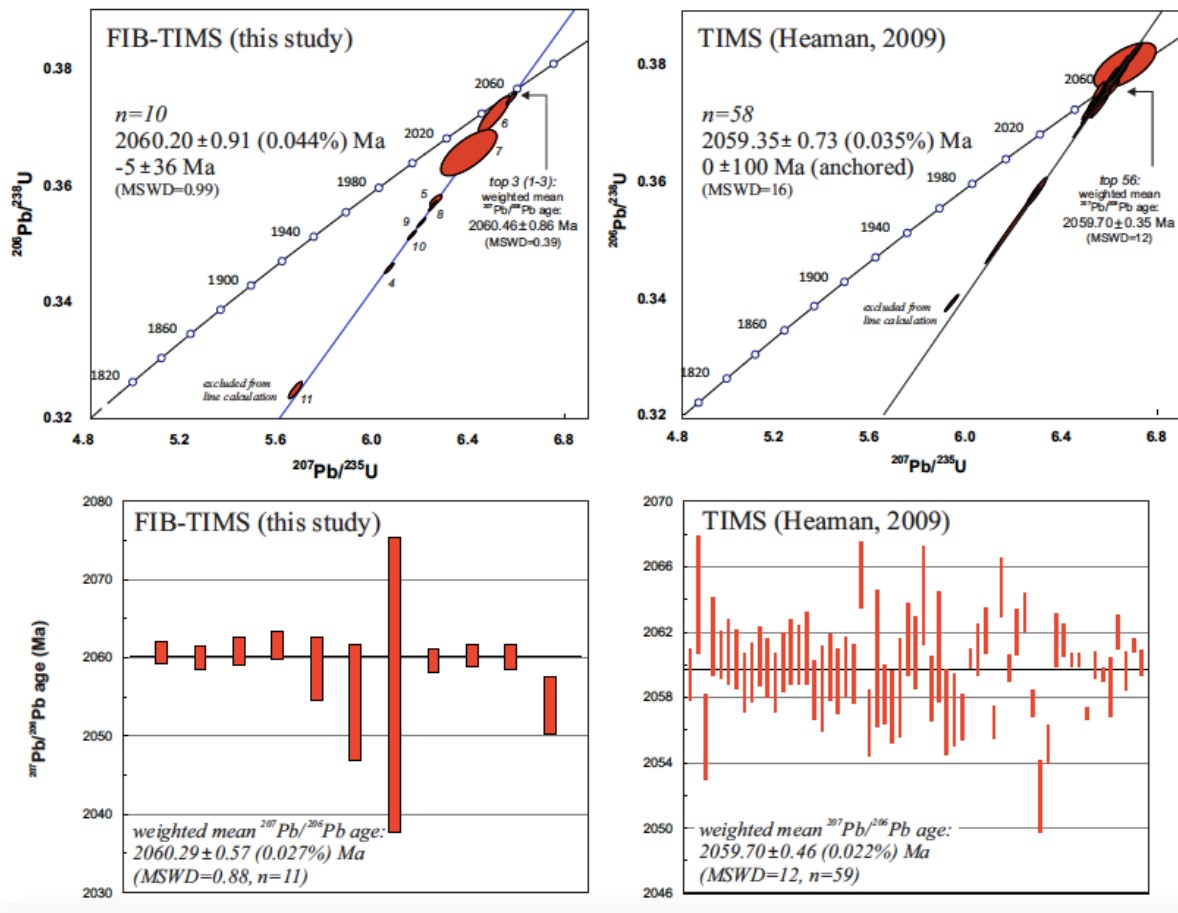


***Figure 3:*** *U-Pb concordia diagrams and weighted average Pb/Pb ages for data generated by*
*FIB-TIMS and TIMS analysis of the Phalaborwa baddeleyite reference material within this*
*study (left). For comparison, all U/Pb and Pb/Pb data reported by Heaman, 2009, are also*
*presented (right), highlighting the natural discordance and variation within the Phalaborwa*
*baddeleyite population. Individual data points are numbered in reference to Table 1.*







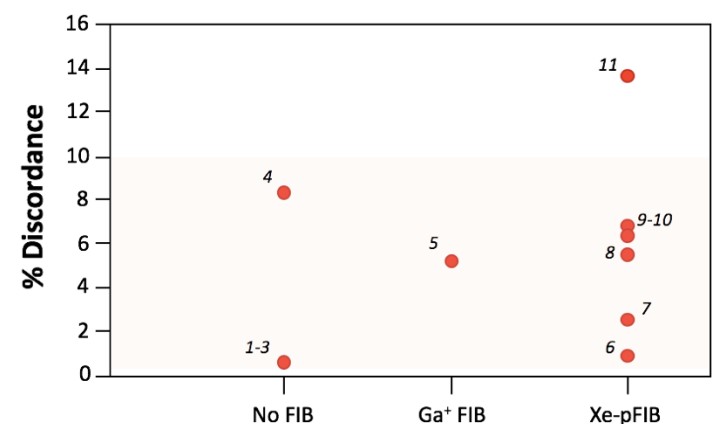

**Figure 4:** *Percentage discordance plotted against extraction method. Discordance reported by Heaman, 2009, is shown by the transparent box, and individual data points are numbered in reference to Table 1.*

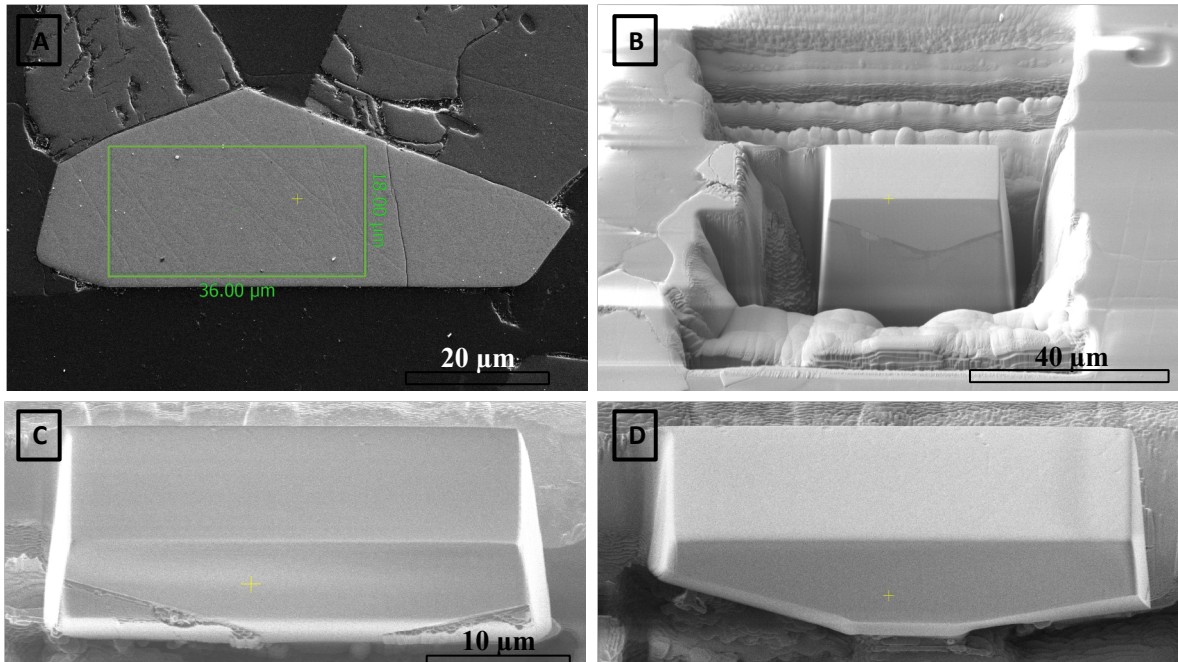


**Figure 5:** *Xe+ pFIB images detailing the extraction of a 36 x 18 μm domain of a large (50*
*μm) baddeleyite grain from a thin section of the Duluth gabbro (**A**). During large scale*
*cutting (**B**), small domains of common-Pb bearing feldspar remained attached to the target*
*baddeleyite (**C**), though these were quickly removed using the Xe+ pFIB instrument through a*
*series of tilted and rotated cuts to produce a single grain with no rim or inclusions (**D**).*