# Peer review of "Highly accurate dating of micrometre-scale baddeleyite domains through combined"

_Geochronology, 2019_

## Referee Comment (RC1) · Graham Edwards (Referee) · 13 Jan 2020

General Comments: White and others demonstrate the potential of coupled FIB extraction and ID-TIMS measurement of U and Pb in baddeleyite to calculate precise Pb-Pb dates of specific baddeleyite domains extracted by FIB from known petrologic contexts. The authors successfully reproduce precise 207Pb/206Pb dates of Phalaborwa complex baddeleyite domains extracted by Xe pFIB milling that are consistent with the 207Pb/206Pb dates of mechanically separated baddeleyite crystals and fragments measured by ID-TIMS in this and other studies (e.g. Heaman, 2009) as well as

by LA-ICP-MS (Ibanez-Mejia et al., 2014). They further show that a single baddeleyite domain milled with a Ga FIB does not deviate noticeably in terms of U-Pb and Pb-Pb systematics from domains milled with a Xe pFIB.

While convincing with regard to Pb-Pb systematics in Phalaborwa baddeleyite, I think the study would benefit from a larger dataset to resolve patterns in U-Pb systematics and be strengthened by testing the methodology in another baddeleyite standard. I am skeptical of the interpretation that FIB-milled baddeleyite domains, as reported in this manuscript, reflect pristine U-Pb systematics relative to their mechanically separated counterparts. However, this skepticism of baddeleyite U-Pb systematics is not limited to the data reported by White and others: U-Pb discordance is a common phenomenon in whole-grain baddeleyite ID-TIMS measurements. Importantly, the present data convincingly support that the Pb-Pb systematics remain unaffected, and I think that this latter finding is most important and ought to be emphasized. Since baddeleyite U-Pb dates are complex and heterogeneous in all contexts, statements that FIB-ID-TIMS U-Pb dates reproduce other ID-TIMS measurements are not necessary to justify the reproducibility and utility of FIB-TIMS methods for baddeleyite Pb-Pb geochronology, and inclusion of these statements would require support from additional data and interpretation.

The study is well-motivated, the data, while limited, is of high quality, and the experimental design, results, and conclusions are generally sound. This paper successfully shows the promise of FIB-ID-TIMS methodologies for application to baddeleyite U-Pb geochronology. This is an exciting advancement that will open the door to calculating dates for new types of samples with petrologic context of measured domains. My opinion is that this manuscript is well-suited to publication in Geochronology if the following issues are appropriately addressed.

Specific Comments:

Methods: 1.) Was a reduction algorithm or software used to calculate dates? When

I calculated discordance following the approach described in line 128 ([100*Pb-Pb age/U-Pb age]-1), I got slightly different values from those reported in Table 1 (e.g. I calculate 4.6% discordance for the 50x50x50 um cube).

2.) In addition to stating what algorithm (if any) is used, the authors should state the assumed 238U/235U composition used in data reduction.

3. Line 184: What was the assumed U mass fractionation and how was it determined? Was 982 used to determine Pb deadtime? How was deadtime determined for U?

4.) The 8N HNO3 wash (line 176) should be explained in more detail, and the selection of this leaching method should be justified. Different leaching techniques have significant effects on U-Pb concordance in baddeleyite. See Rioux et al. (2010, Contrib Mineral Petrol) and discussion below in point 5.

Discussion: 5.) My primary concern with this paper is that the dataset does not convincingly show that the methods applied herein do not perturb the elemental U-Pb systematics in baddeleyite.

The authors state in lines 234-236 "there is no obvious correlation between the severity of discordance and the method used to isolate the domain for TIMS dating." However, with the exception of the "1 chip from mount" fraction, the other three mechanically separated (i.e. not FIB-milled) fractions exhibit very low discordance of ≤0.6%. Thus, of these limited data, 75% are nearly concordant. This is comparable to the cited findings of Heaman (2009): 85% (58 of 68) Phalaborwa baddeleyite fractions are <1% discordant. In contrast, 88% (7 out of 8) FIB-extracted baddeleyite domains exhibit >2% discordance. Thus, there is an apparent correlation between severity of discordance and the method used to isolate the domain for TIMS dating: FIB-extracted baddeleyite domains are more prone to discordance than mechanically isolated domains/grains. This apparent pattern may be a result of the paucity of measurements interpreted (e.g. n=4 mechanically isolated fractions) or reflect perturbance of U-Pb systematics in FIB-extracted samples. Regardless, the statements in lines 234-236 and 239-243 are at

best poorly supported and at worst contradicted by the present data. While the U-Pb system appears perturbed by FIB milling, I agree with the assertion that the Pb-Pb system is not perturbed.

Greater U-Pb discordance in FIB-milled fractions may reflect either a direct effect of the FIB milling or it may reflect the combined effects of FIB milling with the leaching techniques employed here. As addressed above (4), the 8N HNO3 wash prior to spiking and dissolution (line 176) is provided without any justification or more detailed description. However, it has been shown by Rioux and others (2010, Contrib Mineral Petrol) that the U-Pb compositions of baddeleyite grains are sensitive to different chemical abrasion techniques. It may be that the combination of FIB-TIMS and the leaching method employed herein have resulted in the apparent pattern of more prevalent >2% U-Pb discordance in FIB-extracted baddeleyite domains than mechanically separated domains.

Unless the apparent effect on U-Pb systematics is refuted by additional data, I think it is imperative that the authors acknowledge it and make an effort to explain why it might be the case.

6.) While White and others successfully show the preserved Pb-Pb systematics in FIB-milled domains of Phalaborwa baddeleyite, their primary conclusion would be strengthened by confirmation of this behavior in another baddeleyite standard. Figure 4 shows a milled baddeleyite domain from a sample of "Duluth gabbro." Although the specific locality is not stated, the U-Pb systematics of FC-1 and FC-4b baddelyite have apparently been reasonably well characterized by Crowley and Schmitz (2009, AGU Fall Meeting Abstracts), Hoaglund (2010, MSc Thesis), and Schmitt and others (2010, Chem Geology). There may be other studies, these are simply those listed in Ibanez-Mejia and others (2014). If the same sampling and ID-TIMS methodologies applied to Phalaborwa baddeleyite were applied successfully to this Duluth gabbro sample, this would both further strengthen the conclusions AND demonstrate applicability of baddeleyite domain FIB-extraction from within a more complex rock matrix rather than just

from a larger baddeleyite crystal (i.e. Phalaborwa).

7.) In line 192, Pb laboratory blanks are reported at "usually less than 0.5" pg. However, the common Pb mass of a few measurements notably exceed this value, including the "1 chip from mount" (12.29 pg), the "50x50x50 um cube" (1.91 pg), and the "5x15um domain #1" (6.24 pg). Were tpbs measured concurrently and are these Pbc values consistent with those blank amounts? If the values are not consistent, these may reflect portions or domains of the fractions that contain some initial common Pb in addition to laboratory blank contributions. I think the manuscript would benefit from exploring and providing a statement on the sensitivity of the calculated dates to correcting all common Pb as laboratory blank as opposed to applying an initial Pbc correction for Pbc exceeding the measured blank.

8.) Zircon overgrowths/inter-growths are hypothesized to be a contributing factor to discordant U-Pb systematics (e.g. Rioux et al., 2010, Contrib Min Petrol). Lines 283-289 address this process, but do not assess the effect this may have on the present study. Zircon overgrowths are probably unlikely given the extraction of the studied domains from a single baddeleyite crystal, but what about intergrowths/inclusions? Have steps been taken to find and/or control for these? Since HF was used in digestion and would have dissolved any minor zircon domains, I think the manuscript would benefit from investigation of and discussion on whether zircon or other mineral inclusions or intergrowths may be contributing to observed discordance. In terms of the topic of discussion, lines 283-289 fit better in the preceding section 4.2 (Isotopic heterogeneity in Phalaborwa baddeleyite).

Technical Comments (Listed with Line Numbers): 46, 190,210,225: 2s should be $2\sigma$ as in other parts of the paper 94: xenon should begin with a lowercase. 95: special –> spatial? 98: pFIB – I assume the p is for plasma, but should be stated explicitly 132: LA-ICP-MS should be defined before this (e.g. line 87). 167-170: hard to keep track of groupings. I think more punctuation would help e.g. : or -. 506: images –> imaged?

---

## Referee Comment (RC2) · Anonymous Referee #2 · 21 Jan 2020

General comments: White et al. present an interesting new study, demonstrating the potential for using FIB techniques to isolate microdomains of baddeleyite for high precision ID-TIMS U-Pb dating. Such a technique has the potential to be widely applicable to both extraterrestrial and terrestrial samples. The authors demonstrate that FIB microsamples yield both precise and accurate dates for the Phalaborwa baddeleyite standard. This is a timely and important study, complimenting other recent studies on using coupled micro-sampling (e.g., by laser cutting) and TIMS analyses to obtain precise and accurate dates with high spatial resolution.

[Figure]

Specific comments: Lines 187–191: The authors argue that all common Pb (Pbc) in the samples comes from laboratory blank, and use an estimate of the laboratory blank isotopic composition to correct their analyses. Given the large range of observed Pbc (0.17–12.29 pg), it seems likely that either 1. not all of the Pbc is laboratory blank and some Pbc is coming from within the grains or 2. the analytical blank is highly variable. I suspect the former is likely true, in which case, some of the Pbc should be accounted for as initial Pb, with a different isotopic composition than the laboratory blank. If the latter is true, this raises questions about how representative the applied blank isotopic composition is; the blank isotopic compositions were likely measured on better behaved total procedural blanks, while the highly variable Pbc of these analyses suggest a range of different blank sources, potentially related to a mix of reagent blanks, sample handling and other factors.

The uncertainties on the applied blank isotopic compositions also seem low to me. It has been common in the ID-TIMS community to assume relatively low uncertainty in the blank isotopic composition; however, repeat measurements of total procedural blank isotopic compositions at the MIT, Boise State and Princeton U-Pb labs have all found uncertainties in the blank isotopic compositions that are approximately an order of magnitude higher than those used in this study ($\sim$3–4 %; e.g. Schoene et al., Science, 2019, supplemental material). The low assumed uncertainties in the blank isotopic composition in this study are especially questionable given the large range in Pbc observed in the dated grains, as discussed above.

The authors should outline how the blank isotopic compositions and uncertainties were determined. If they are not based on the measured isotopic composition and variability of total procedural blanks, it would be worth measuring a series of blanks. At a minimum, the authors should discuss the impact that variable blank isotopic compositions would have on the calculated dates and uncertainties. Given the low 206Pb/204Pb of some analyses, some of the data will be sensitive to the blank parameters.

Lines 219–222 and lines 239–243: The authors argue that the lack of correlation between discordance and FIB exposure time provide strong evidence that the FIB does not lead to Pb movement or Pb-loss. This does not seem like a robust conclusion. As the authors point out, there is significant scatter and variable discordance in previous analyses of untreated Phalaborwa grains and grain fragments by Heaman and others. Given this natural variability, the lack of a correlation between discordance and FIB exposure is not meaningful. It may be that there is significant FIB induced Pb-loss, but because it is being superimposed on the natural Phalaborwa variability, it does not lead to a clear correlation. For example, it is possible that a sample with minimal FIB exposure was naturally discordant, while a sample with extensive FIB exposure was originally concordant, but the FIB exposure led to a discordant date. These data would not show a correlation between exposure time and discordance, even though the FIB did lead to Pb-loss.

I doubt that the FIB does induce significant Pb-loss, but the current dataset does not provide an adequate test of this. The authors point out that the smallest FIB sample yielded the most discordant date, which does raise concerns. It would be interesting to do a similar experiment on a sample with consistently concordant baddeleyite, such that any FIB induced discordance could be resolved. It would also be interesting to do either SIMS or laser depth profiling of FIB extracted microsamples. While the precision of these techniques is lower, they might reveal any FIB induced effects along the sampled domain margins.

Lines 291–292:

The authors suggest that FIB TIMS analyses will be limited by counting statistics for small samples. While this will depend on the U content and age, in many cases, sample size is likely to be limited by uncertainty in the laboratory blank isotopic composition, rather than counting statistics. At low Pb, the uncertainty from the Pb blank will dominate the total uncertainty. It would be interesting for the authors to model how sample size will be limited by age and U content, in order to provide a more general conclusion on the minimum possible sample sizes for future work.

Technical corrections: Line 51: I would recommend adding a comma after "ID-TIMS".

Line 62: It would be good to add a reference to Rioux et al., CMP, 2010 after Krogh, 1982. Rioux et al. did extensive experiments on chemical abrasion of baddeleyite.

Line 63: I would recommend adding a comma after "minerals".

Line 71: I would recommend adding a comma after "...Darling et al., 2016)".

Line 177: It would be useful to indicate that the grains were dissolved in Parr acid digestion vessels.

Lines 192–193: "Routine testing indicates that laboratory blanks for Pb and U are usually less than 0.5 and 0.01 pg, respectively, but common Pb can be introduced during analysis." This sentence is vague. The authors should specify where they suspect the excess Pbc is coming from. As I discussed above, it seems plausible it is initial Pbc from the grains.

Line 198: "is" should be replaced with "are".

Line 201: "analysis" should be replaced with "analyses".

Line 205: I would recommend adding a comma after "(Heaman, 2009)".

Line 237 and line 270: It would be clearer to indicate which U-Pb dates the authors are referring to (i.e. 206Pb/238U or 207Pb/235U).

Lines 283–285: "An additional possible source of discordance in baddeleyite U-Pb TIMS analysis is the incorporation of zircon overgrowths (which are subjected to Pb loss; Pietrzak-Renaud and Davis, 2014)". The authors should add a reference to Davidson and van Breemen (CMP, 1998), who did extensive work on zircon overgrowths of baddeleyite.

Table 1: Why not just include the full data table? It is only 1 page long and would not take up much more space than the summary table.

**GChronD**

---

## Referee Comment (RC3) · Joshua Davies (Referee) · 30 Jan 2020

I apologise for the lateness of my review...!

The technique is supposed to be combining insitu analysis with TIMS, however the only extraction from a thin section is baddeleyite from the Duluth gabbro that is subsequently not analysed – why is this?. The authors indicate how much better this technique is for characterizing the material before TIMS dating however they do not characterize their material before TIMS dating... there is no chemical mapping, no idea of where

the samples are taken from in the baddeleyite crystals, no real electron imaging which seems strange when the authors indicate how crucial these techniques are before TIMS measurements.

Also, the figures are not really helping the manuscript much. Figure 3 is not particularly useful. It doesn't show the differences between the extraction methods, nor the amount of Pb in each sample, or the common Pb. It should highlight the scale difference between the Heaman weighted mean data and yours as well. I would like to see a figure relating each extraction technique with the common Pb or discordance or something like that to show the potential impact of the tecniques on the data. I suspect that the dataset isn't really large enough to ascertain whether the extraction techniques have an impact on common Pb etc, but the reader can't tell from the current figures.

Just a final small point about the common Pb issues. This technique is designed to extract well characterized pristine sections of baddeleyite grains, and therefore I guess the extracted grain fragments/ areas of the phalaborwa crystals were chosen for their general inclusion free nature? Can we assume that the high common Pb found in some of the analysis is either from the handling associated with the extraction or from the lab? Or from inclusions that weren't identified prior to FIB extraction? The authors should at least comment on this aspect of their data and explain how they corrected for different sources of common Pb – if they expect sources other than just lab blank.

Line by line comments –

Line 46 – $\sim 0.1\%$ 2s - change 2s to $2\sigma$ (or indicate what you mean by 's')

Line 55 – it is not true that grains can not be characterized prior to TIMS work. There are plenty of papers which perform electron microscopy, trace element analysis by LA-ICP-MS, oxygen isotopes etc before TIMS work on the same grains – See Farina et al. 2018 (EPSL), Barboni et al. 2018 (Science advances) for a couple of examples. There are many labs, which routinely perform 'insitu' (in grain mounts) analysis before TIMS dating. I believe you are referring only to the petrological context of grains, which is
lost by making grain mounts. Also, the (Paquette et al. 2004) reference would be good here.

Line 62 – Again the text is correct but a little misleading – TIMS dating can not remove different fragments of grains or chose areas of grains to date, however the geochronologist can break grains into fragments and date different zones of zircon grains for example in Reimink et al. 2016 (Nature Geoscience) we chemically characterized zircon growth domains and then performed chemical abrasion on a lot of zircon crystals. We re-measured the zircon fragments after chemical abrasion to identify the different growth domains before TIMS U-Pb geochronology. Also in Gordon et al. 2010 (GSA Bulletin) they broke zircon crystals into different domains before dating. I appreciate that your technique is a significant advancement, but the historical literature should be referenced.

Line 87 – you indicate that laser cutting would induce localized elemental fractionation – would this be avoided by using a fempto second laser?

Line 125 – Heaman 2009, all of the ages in the Heaman 2009 study come from one single baddeleyite megacryst, are the grains measured in this current work from the same megacryst? It's not exactly clear from what you say.

Line 127 – (58/68) – this needs correcting

Line 128 – I guess this is for the editors of the journal, but its likely that you don't need to explain the discordance calculation, if you want to do it, put the proper calculation.

Line 135 – This is slightly confusing wording for the readers and it is also related to problems using discordance as a metric. You mention that 77% of the analyses by Ibanez-Mejia are >1% discordant, but are the errors on these analysis overlapping with the Concordia curve?

Line 187 – how did you measure mass fractionation for each cycle on the mass spectrometer if you didn't use a double spike?

Line 195 – I'm sure this doesn't make a difference to the end results – but why did you assume an average crystal Th/U of the magma since these crystals are not exactly from "an average" magma composition.

Line 231 – you say that the effects of FIB extraction on U-Pb isotope systematics have never been explored – what about atom probe analysis? (the lead author of this work previously published a paper on baddeleyite U-Pb systematics on samples extracted by FIB techniques and then measured on the atom probe)

Lines 236-238 – it would be much better to show graphically the results here rather than explaining them. Ideally you would have a graph that shows age/discordance/common Pb or some other metrics vs the different methods used. Currently it is not entirely convincing that the extraction techniques do not induce some U-Pb disturbance.

Section 4.2 – isotopic heterogeneity in Phalaborwa baddeleyite. It is known that the Phalaborwa baddeleyite grains contain some amount of Pb loss (and this may be endemic to baddeleyite in general – see Schaltegger and Davies 2018, Davis and Davis 2017). Therefore it seems a bit counter intuitive to use the U/Pb ratio of crystals to discuss isotopic heterogeneity. Its not clear if your discussion here is related to Pb loss or real age variation? I think this section needs to be reworded to make it more clear in that regard. Also, the Phalaborwa baddeleyite is a reference material not a standard (and you should refer to it as such throughout the paper). You also mention some degree of "care" (line 263) that should be taken when doing small volume U-Pb work on phalaborwa. What exactly do you mean by this? Do you have a recipe that should be followed to ensure that the best measurements can be made – this would be an interesting and useful addition to the paper.

Lines 269-281 – You cite the work of Davis and Davis 2017, which discusses alpha recoil effects on baddeleyite ages. This current study could have been the perfect test case for the idea of Davis and Davis since you could have dated at high precision areas of baddeleyite crystals at the rim and at the centre of a large crystal. There

are also some confusing sentences in this paragraph that should be corrected slightly. For example, the last sentence says – "allowing the targeted extraction of centralized regions which are unlikely to have lost Pb during an alpha recoil event" - the central regions of grains will have experienced Pb redistribution due to alpha recoil, but will not have lost Pb since adjacent areas will have ejected their Pb into the central region. I think you mean that there will have been no alpha recoil ejection from the centre of the grain.

Line 285 – add Rioux et al. 2010 (Contributions Mineralogy Petrology) here, you can also add this in the introduction since it's an important piece of work on baddeleyite U-Pb geochronology

Lines 291 to 295 – the other reviewers raised concerns about the common Pb in some of the analysis here and their questions adequately covered my concerns.

Lines 296-299 – this is because you are doing TIMS analysis – I'm not sure that it is relevant to say that certain problems only associated with SIMS analysis are avoided by this technique – you are not avoiding these problems because these problems are not associated with your technique.

---

## Author Response (AR1)

**Dr. Lee Francis White**
Department of Natural History
Royal Ontario Museum
Queens Park, Toronto
ON M5S 2C6, Canada
T: 647-230-4212
E: lwhite@rom.on.ca

23rd April 2020

**RE**: gchron-2019-17 "Highly accurate dating of micrometer-scale baddeleyite domains through combined focused ion beam extraction and U-Pb thermal ionization mass spectrometry (FIB-TIMS)"

Dear Brenhin,

Please find attached updated documents for the resubmission of our *Geochronology* article, as detailed above. Below we detail our responses to the reviewer's points:

**Reviewer 1 (Graham Edwards)**

General Comments: White and others demonstrate the potential of coupled FIB ex- traction and ID-TIMS measurement of U and Pb in baddeleyite to calculate precise Pb-Pb dates of specific baddeleyite domains extracted by FIB from known petrologic contexts. The authors successfully reproduce precise 207Pb/206Pb dates of Phalaborwa complex baddeleyite domains extracted by Xe pFIB milling that are consistent with the 207Pb/206Pb dates of mechanically separated baddeleyite crystals and fragments measured by ID-TIMS in this and other studies (e.g. Heaman, 2009) as well as by LA-ICP-MS (Ibanez-Mejia et al., 2014). They further show that a single baddeleyite domain milled with a Ga FIB does not deviate noticeably in terms of U-Pb and Pb-Pb systematics from domains milled with a Xe pFIB.

While convincing with regard to Pb-Pb systematics in Phalaborwa baddeleyite, I think the study would benefit from a larger dataset to resolve patterns in U-Pb systematics and be strengthened by testing the methodology in another baddeleyite standard. I am skeptical of the interpretation that FIB-milled baddeleyite domains, as reported in this manuscript, reflect pristine U-Pb systematics relative to their mechanically separated counterparts. However, this skepticism of baddeleyite U-Pb systematics is not limited to the data reported by White and others: U-Pb discordance is a common phenomenon in whole-grain baddeleyite ID-TIMS measurements. Importantly, the present data convincingly support that the Pb-Pb systematics remain unaffected, and I think that this latter finding is most important and ought to be emphasized. Since baddeleyite U-Pb dates are complex and heterogeneous in all contexts, statements that FIB-ID-TIMS U-Pb dates reproduce other ID-TIMS measurements are not necessary to justify the reproducibility and utility of FIB-TIMS methods for baddeleyite Pb-Pb geochronology, and inclusion of these statements would require support from additional data and interpretation.

The study is well-motivated, the data, while limited, is of high quality, and the experimental design, results, and conclusions are generally sound. This paper successfully shows the promise of FIB-ID-TIMS methodologies for application to baddeleyite U-Pb geochronology. This is an exciting advancement that will open the door to calculating dates for new types of samples with petrologic context of measured domains. My opinion is that this manuscript is well-suited to publication in Geochronology if the following issues are appropriately addressed.

We thank the reviewer for the supportive comments regarding the accurate measurement of Pb-Pb ratios, as well the importance of the technique as an advance in TIMS geochronology. We agree that, due to the inherent discordance of U-Pb systematics in whole-grain ID-TIMS measurements, the observation of discordance here is less important than the reproducible Pb-Pb ratios we report. We have changed the focus of the discussion to better reflect this aspect of the dataset. We also thank the reviewer for the detailed comments below and hope that our responses satisfy any and all concerns with the manuscript.

Specific Comments:
Methods: 1.) Was a reduction algorithm or software used to calculate dates? When I calculated discordance following the approach described in line 128 ([100*Pb-Pb age/U-Pb age]-1), I got slightly different values from those reported in Table 1 (e.g. I calculate 4.6% discordance for the 50x50x50 um cube).
We use: 100*[1 - {(206Pb/238U)/(EXP(L238*T)-1)}] where T is the 7/6 age (and curly bracket factor is measured 6/8 ratio over what it would be for a point on concordia at the 7/6 age).

2.) In addition to stating what algorithm (if any) is used, the authors should state the assumed 238U/235U composition used in data reduction.
We have re-processed the data using the Hiess et al. value (137.818). This is included in the Table footnotes and has been added to the analytical methods section (the data in our original submission used the Steiger and Jager 1977 value of 137.88)

3. Line 184: What was the assumed U mass fractionation and how was it determined? Was 982 used to determine Pb deadtime? How was deadtime determined for U?

**A U mass fractionation of 0.1% per atomic mass unit was used. The deadtime for Pb was determined using SRM 982, and U500 was used for determining U dead time (and now noted in the analytical methods section).**

4.) The 8N HNO3 wash (line 176) should be explained in more detail, and the selection of this leaching method should be justified. Different leaching techniques have significant effects on U-Pb concordance in baddeleyite. See Rioux et al. (2010, Contrib Mineral Petrol) and discussion below in point 5.

**HNO3 is not used to "leach" the crystals, but merely to remove surface Pb prior to analysis. Text has been added to describe how the grains were washed in room temperature HNO3 on the surface of clean parafilm using a micropipette. Rioux et al. note on p. 18 "the paired analyses of HNO3-rinsed and H2O rinsed grain fragments suggest that the low-temperature HNO3 fluxing prior to digestion did not have a significant impact on U-Pb dates." The entire paragraph that follows reinforces the idea that a brief wash at low temperatures is unlikely to have an effect on U-Pb systematics. Further, tests in our laboratory (unpublished) show that even in hot 80ºC concentrated HNO3 for many hours, the impact on systematics is negligible.**

Discussion: 5.) My primary concern with this paper is that the dataset does not convincingly show that the methods applied herein do not perturb the elemental U-Pb systematics in baddeleyite.

The authors state in lines 234-236 "there is no obvious correlation between the severity of discordance and the method used to isolate the domain for TIMS dating." However, with the exception of the "1 chip from mount" fraction, the other three mechanically separated (i.e. not FIB-milled) fractions exhibit very low discordance of ≤0.6%. Thus, of these limited data, 75% are nearly concordant. This is comparable to the cited findings of Heaman (2009): 85% (58 of 68) Phalaborwa baddeleyite fractions are <1% discordant. In contrast, 88% (7 out of 8) FIB-extracted baddeleyite domains exhibit >2% discordance. Thus, there is an apparent correlation between severity of discordance and the method used to isolate the domain for TIMS dating: FIB-extracted baddeleyite domains are more prone to discordance than mechanically isolated domains/grains. This apparent pattern may be a result of the paucity of measurements interpreted (e.g. n=4 mechanically isolated fractions) or reflect perturbance of U-Pb systematics in FIB- extracted samples. Regardless, the statements in lines 234-236 and 239-243 are at best poorly supported and at worst contradicted by the present data. While the U-Pb system appears perturbed by FIB milling, I agree with the assertion that the Pb-Pb system is not perturbed.

Greater U-Pb discordance in FIB-milled fractions may reflect either a direct effect of the FIB milling or it may reflect the combined effects of FIB milling with the leaching techniques employed here. As addressed above (4), the 8N HNO3 wash prior to spiking and dissolution (line 176) is provided without any justification or more detailed description. However, it has been shown by Rioux and others (2010, Contrib Mineral Petrol) that the U-Pb compositions of baddeleyite grains are sensitive to different chemical abrasion techniques. It may be that the combination of FIB-TIMS and the leaching method employed herein have resulted in the apparent pattern of more prevalent >2% U-Pb discordance in FIB-extracted baddeleyite domains than mechanically separated domains.

Unless the apparent effect on U-Pb systematics is refuted by additional data, I think it is imperative that the authors acknowledge it and make an effort to explain why it might be the case.

**We agree with the reviewer's suggestion that we better highlight the unperturbed Pb-Pb isotope data. We have now included a more complete discussion on this within the manuscript and have also reworded the closing lines of the abstract to better reflect the unresolved source of U-Pb discordance and better highlight the reproducible Pb-Pb ages generated by the FIB-TIMS method. As discussed above, we note that the HN03 wash would not affect the measured U-Pb systematics of the baddeleyite analysed here. In addition, the minimal extent of damage induced by the Xe-pFIB (nm-scale depths; Burnett et al., 2016) could not account for the more discordant data points (13.6% discordance) even if coupled with Pb loss through leaching. These points have now been better highlighted within the paper.**

6.) While White and others successfully show the preserved Pb-Pb systematics in FIB- milled domains of Phalaborwa baddeleyite, their primary conclusion would be strengthened by confirmation of this behavior in another baddeleyite standard. Figure 4 shows a milled baddeleyite domain from a sample of "Duluth gabbro." Although the specific locality is not stated, the U-Pb systematics of FC-1 and FC-4b baddeleyite have apparently been reasonably well characterized by Crowley and Schmitz (2009, AGU Fall Meeting Abstracts), Hoaglund (2010, MSc Thesis), and Schmitt and others (2010, Chem Geology). There may be other studies, these are simply those listed in Ibanez- Mejia and others (2014). If the same sampling and ID-TIMS methodologies applied to Phalaborwa baddeleyite were applied successfully to this Duluth gabbro sample, this would both further strengthen the conclusions AND demonstrate applicability of baddeleyite domain FIB-extraction from within a more complex rock matrix rather than just from a larger baddeleyite crystal (i.e. Phalaborwa).

**While we agree with the reviewer, unfortunately the grain was lost during extraction from the grain mount. Nonetheless, we hope that by demonstrating the utility of the FIB-TIMS technique, further work by the community (both sub-sampling crystal domains and extracting grains directly from thin section) will reinforce the potential applications of the technique.**

7.) In line 192, Pb laboratory blanks are reported at "usually less than 0.5" pg. However, the common Pb mass of a few measurements notably exceed this value, including the "1 chip from mount" (12.29 pg), the "50x50x50 um cube" (1.91 pg), and the "5x15um domain #1" (6.24 pg). Were tpbs measured concurrently and are these Pbc values consistent with those blank amounts? If the values are not consistent, these may reflect portions or domains of the fractions that contain some initial common Pb in addition to laboratory blank contributions. I think the manuscript would benefit from exploring and providing a statement on the sensitivity of the calculated dates to correcting all common Pb as laboratory blank as opposed to applying an initial Pbc correction for Pbc exceeding the measured blank.

**Agreed—we have added text to address this, including recalculations to test the sensitivity to different Pb isotopic compositions. Of course, it is not possible to know whether the source is geological or from the laboratory, however, whether model Pb ratios from Stacey and Kramer's or a depleted mantle Pb composition is used above an assumed 0.5 pg blank level, the "1 chip from mount (12.29 pg)" or the "50x50x50 um cube (1.91 pg), it does not impact the age a significant amount (described in text).**

8.) Zircon overgrowths/inter-growths are hypothesized to be a contributing factor to discordant U-Pb systematics (e.g. Rioux et al., 2010, Contrib Min Petrol). Lines 283- 289 address this process, but do not assess the effect this may have on the present study. Zircon overgrowths are probably unlikely given the extraction of the studied domains from a single baddeleyite crystal, but what about intergrowths/inclusions? Have steps been taken to find and/or control for these? Since HF was used in digestion and would have dissolved any minor zircon domains, I think the manuscript would benefit from investigation of and discussion on whether zircon or other mineral inclusions or intergrowths may be contributing to observed discordance. In terms of the topic of discussion, lines 283-289 fit better in the preceding section 4.2 (Isotopic heterogeneity in Phalaborwa baddeleyite).

**Imaging of the grain prior to FIB work (reflected light and backscatter & secondary electron) allowed for characterisation of the chosen lift out areas. For all domains, cracks and inclusions were avoided (e.g. Figure 2) to prevent domains that may have experienced Pb loss being incorporated into the TIMS data point. Thus, it is highly unlikely that zircon (or other mineral inclusions) are responsible for the observed discordance given the targeted nature of the FIB-TIMS technique. We have moved the text on lines 283-289 to fit within section 4.2 (Isotopic heterogeneity) as suggested.**

**Line-by-line comments**
46, 190,210,225: 2s should be 2σ as in other parts of the paper **Done**
94: xenon should begin with a lowercase. **Done**
95: special –> spatial? **Good catch, and done**
98: pFIB – I assume the p is for plasma, but should be stated explicitly 132: LA-ICP-MS should be defined before this (e.g. line 87) **Done on L87**.
167-170: hard to keep track of groupings. I think more punctuation would help e.g. : or -. **Done**
506: images –> imaged? **Imaged is correct in this case.**

**Reviewer 2 (Anonymous)**
General comments: White et al. present an interesting new study, demonstrating the potential for using FIB techniques to isolate microdomains of baddeleyite for high precision ID-TIMS U-Pb dating. Such a technique has the potential to be widely applicable to both extraterrestrial and terrestrial samples. The authors demonstrate that FIB microsamples yield both precise and accurate dates for the Phalaborwa baddeleyite standard. This is a timely and important study, complimenting other recent studies on using coupled micro-sampling (e.g., by laser cutting) and TIMS analyses to obtain precise and accurate dates with high spatial resolution. **We thank the reviewer for the supportive comments and hope we have suitably addressed any and all concerns raised in the following line-by-line comments.**

Specific comments: Lines 187–191: The authors argue that all common Pb (Pbc) in the samples comes from laboratory blank, and use an estimate of the laboratory blank isotopic composition to correct their analyses. Given the large range of observed Pbc (0.17–12.29 pg), it seems likely that either 1. not all of the Pbc is laboratory blank and some Pbc is coming from within the grains or 2. the analytical blank is highly variable. I suspect the former is likely true, in which case, some of the Pbc should be accounted for as initial Pb, with a different isotopic composition than the laboratory blank. If the latter is true, this raises questions about how representative the applied blank isotopic composition is; the blank isotopic compositions were likely measured on better behaved total procedural blanks, while the highly variable Pbc of these analyses suggest a range of different blank sources, potentially related to a mix of reagent blanks, sample handling and other factors.

**We have addressed this in our response to comments by reviewer 1. Given we have no evidence of Pb residing within the baddeleyite, i.e. in micro-inclusions or fracture-hosted alteration zones, we tend to think that analytical sources may the cause. It should also be noted that the absolute amount of common Pb has less to do with the potential effect on ratios than the ratio of radiogenic Pb to common Pb does. Nevertheless, the reviewer is correct in suggesting that geological sources are possible. We have therefore reprocessed data for several of the results with total common Pbs of >1 pg (i.e., 12.29 pg, 1.91 pg)**

The uncertainties on the applied blank isotopic compositions also seem low to me. It has been common in the ID-TIMS community to assume relatively low uncertainty in the blank isotopic composition; however, repeat measurements of total procedural blank isotopic compositions at the MIT, Boise State and Princeton U-Pb labs have all found uncertainties in the blank isotopic compositions that are approximately an order of magnitude higher than those used in this study (∼3–4 %; e.g. Schoene et al., Science, 2019, supplemental material). The low assumed uncertainties in the blank isotopic composition in this study are especially questionable given the large range in Pbc observed in the dated grains, as discussed above.

**All U-Pb analytical data uncertainties have been generated with blank isotopic uncertainties of 4% and this has been corrected in the text.**

The authors should outline how the blank isotopic compositions and uncertainties were determined. If they are not based on the measured isotopic composition and variability of total procedural blanks, it would be worth measuring a series of blanks. At a minimum, the authors should discuss the impact that variable blank isotopic compositions would have on the calculated dates and uncertainties. Given the low 206Pb/204Pb of some analyses, some of the data will be sensitive to the blank parameters.

**As above, all data generated with 4% uncertainties.**

Lines 219–222 and lines 239–243: The authors argue that the lack of correlation between discordance and FIB exposure time provide strong evidence that the FIB does not lead to Pb movement or Pb-loss. This does not seem like a robust conclusion. As the authors point out, there is significant scatter and variable discordance in previous analyses of untreated Phalaborwa grains and grain fragments by Heaman and others. Given this natural variability, the lack of a correlation between discordance and FIB exposure is not meaningful. It may be that there is significant FIB induced Pb-loss, but because it is being superimposed on the natural Phalaborwa variability, it does not lead to a clear correlation. For example, it is possible that a sample with minimal FIB exposure was naturally discordant, while a sample with extensive FIB exposure was originally concordant, but the FIB exposure led to a discordant date. These data would not show a correlation between exposure time and discordance, even though the FIB did lead to Pb-loss.

I doubt that the FIB does induce significant Pb-loss, but the current dataset does not provide an adequate test of this. The authors point out that the smallest FIB sample yielded the most discordant date, which does raise concerns. It would be interesting to do a similar experiment on a sample with consistently concordant baddeleyite, such that any FIB induced discordance could be resolved. It would also be interesting to do either SIMS or laser depth profiling of FIB extracted microsamples. While the precision of these techniques is lower, they might reveal any FIB induced effects along the sampled domain margins.

**We thank the reviewer for the comment and agree that the discordance of U-Pb systematics in Phalaborwa, as observed here, cannot be wholly ascribed to either FIB interaction, natural variance within the grain, or a combination of the two processes. However, we have now strengthened discussion on the absence of Pb diffusion during FIB preparation of atom probe tip specimen, which, though operated at a lower accelerating voltage than a Xe-pFIB, is expected to induce more damage than a Ga-source instrument (Burnett et al., 2014). This discussion further supports our interpretation that the Xe-pFIB would not induce localised damage nor isotopic mobility within the extracted mass, and that instead the observed discordance can be confidently associated with natural variation within the grain. In addition, by addressing an additional comment from reviewer #1, we have also altered the concluding statements within the abstract, which now focus on the precise and reproducible Pb-Pb data generated by the FIB-TIMS technique and less on the discordant U-Pb data.**

Lines 291–292: The authors suggest that FIB TIMS analyses will be limited by counting statistics for small samples. While this will depend on the U content and age, in many cases, sample size is likely to be limited by uncertainty in the laboratory blank isotopic composition, rather than counting statistics. At low Pb, the uncertainty from the Pb blank will dominate the total uncertainty. It would be interesting for the authors to model how sample size will be limited by age and U content, in order to provide a more general conclusion on the minimum possible sample sizes for future work.

**We agree that this would be interesting, but any such model would have to include a wide range of variables that are beyond the scope of this study. We also note (both here and within the manuscript) that at the smallest sizes (<15 um) sample manipulation (extraction and dissolution) will be the primary challenge.**

**Line-by-line comments**
Line 51: I would recommend adding a comma after "ID-TIMS". **Done.**

Line 62: It would be good to add a reference to Rioux et al., CMP, 2010 after Krogh, 1982. Rioux et al. did extensive experiments on chemical abrasion of baddeleyite. **Added.**

Line 63: I would recommend adding a comma after "minerals".
Line 71: I would recommend adding a comma after ". . .Darling et al., 2016)". **Done.**

Line 177: It would be useful to indicate that the grains were dissolved in Parr acid digestion vessels. **A more complete methodology is now presented in section 2.2.**

Lines 192–193: "Routine testing indicates that laboratory blanks for Pb and U are usually less than 0.5 and 0.01 pg, respectively, but common Pb can be introduced during analysis." This sentence is vague. The authors should specify where they suspect the excess Pbc is coming from. As I discussed above, it seems plausible it is initial Pbc from the grains.

**It is not possible to know whether excess Pb was introduced during lab procedures or resided in the crystal. As in zircon, Pb is unlikely to come from within the baddeleyite crystal lattice owing to its too large ionic radius, but could be introduced to the grain in small inclusions or from alteration in fractures (none was observed using SEM and optical microscopy). As described in the text, and above, sensitivity tests indicate crustal and mantle compositions for the common Pb in excess of the assumed 0.5 pg blank is negligible to the age calculations.**

Line 198: "is" should be replaced with "are". **Done.**

Line 201: "analysis" should be replaced with "analyses". **Done.**

Line 205: I would recommend adding a comma after "(Heaman, 2009)". **Done.**

Line 237 and line 270: It would be clearer to indicate which U-Pb dates the authors are referring to (i.e. 206Pb/238U or 207Pb/235U). **Corrected to properly reference $^{206}Pb/^{238}U$.**

Lines 283–285: "An additional possible source of discordance in baddeleyite U-Pb TIMS analysis is the incorporation of zircon overgrowths (which are subjected to Pb loss; Pietrzak-Renaud and Davis, 2014)". The authors should add a reference to Davidson and van Breemen (CMP, 1998), who did extensive work on zircon overgrowths of baddeleyite. **Done.**

Table 1: Why not just include the full data table? It is only 1 page long and would not take up much more space than the summary table.
**Included**

**Reviewer 3 (Joshua Davies)**

The technique is supposed to be combining in-situ analysis with TIMS, however the only extraction from a thin section is baddeleyite from the Duluth gabbro that is subsequently not analysed – why is this?. The authors indicate how much better this technique is for characterizing the material before TIMS dating however they do not characterize their material before TIMS dating. . . there is no chemical mapping, no idea of where  the samples are taken from in the baddeleyite crystals, no real electron imaging which seems strange when the authors indicate how crucial these techniques are before TIMS measurements.
**For the large baddeleyite grain analysed here, extensive pre-characterisation would have been an expensive and time-consuming step that would have added little to the paper. Previous EBSD work on a similar grain, for example (as reported in White et al., 2018, Geology) reveals a complete absence of internal structure and complexity. In comparison, meteoritic baddeleyite show a wide range of twin and recrystallization relationships that are shown to have a direct effect on the measured U-Pb systematics (e.g. Darling et al., 2016, EPSL; White et al., 2019, Geoscience Frontiers). For these precious materials pre-characterisation will be key, but such integrated studies are clearly beyond the scope of this methodological paper.**

Also, the figures are not really helping the manuscript much. Figure 3 is not particularly useful. It doesn't show the differences between the extraction methods, nor the amount of Pb in each sample, or the common Pb. It should highlight the scale difference between the Heaman weighted mean data and yours as well. I would like to see a figure relating each extraction technique with the common Pb or discordance or something like that to show the potential impact of the tecniques on the data. I suspect that the dataset isn't really large enough to ascertain whether the extraction techniques have an impact on common Pb etc, but the reader can't tell from the current figures.
**We have recreated Figure 3 to include numbered references to Table 1, allowing the reader to unambiguously link data points on the concordia plots to the extraction technique implemented in sample preparation. We have also included a new figure (Figure 4) which shows extraction technique (no FIB, Ga⁺ FIB or Xe-pFIB) compared to measured % discordance. Again, these data points are numbered to allow direct comparison to Table 1.**

Just a final small point about the common Pb issues. This technique is designed to extract well characterized pristine sections of baddeleyite grains, and therefore I guess the extracted grain fragments/ areas of the phalaborwa crystals were chosen for their general inclusion free nature? Can we assume that the high common Pb found in some of the analysis is either from the handling associated with the extraction or from the lab? Or from inclusions that weren't identified prior to FIB extraction? The authors should at least comment on this aspect of their data and explain how they corrected for different sources of common Pb – if they expect sources other than just lab blank.
**We have updated the description of the sources of and correction for common-Pb in the manuscript. Please see our responses to other reviewers for details.**

**Line by line comments**
Line 46 – ∼0.1% 2s - change 2s to 2σ (or indicate what you mean by 's') **Done.**

Line 55 – it is not true that grains can not be characterized prior to TIMS work. There are plenty of papers which perform electron microscopy, trace element analysis by LA- ICP-MS, oxygen isotopes etc before TIMS work on the same grains – See Farina et al. 2018 (EPSL), Barboni et al. 2018 (Science advances) for a couple of examples. There are many labs, which routinely perform 'insitu' (in grain mounts) analysis before TIMS dating. I believe you are referring only to the petrological context of grains, which is lost by making grain mounts. Also, the (Paquette et al. 2004) reference would be good here.
**We have reworded this statement to read: *"As a result, the analysed grains preserve no evidence for their petrological or mineralogical context and are incredibly challenging to characterise (e.g. electron microscopy) prior to dating…."***

Line 62 – Again the text is correct but a little misleading – TIMS dating can not remove different fragments of grains or chose areas of grains to date, however the geochronologist can break grains into fragments and date different zones of zircon grains for example in Reimink et al. 2016 (Nature Geoscience) we chemically characterized zircon growth domains and then performed chemical abrasion on a lot of zircon crystals. We re-measured the zircon fragments after chemical abrasion to identify the different growth domains before TIMS U-Pb geochronology. Also in Gordon et al. 2010 (GSA Bulletin) they broke zircon crystals into different domains before dating. I appreciate that your technique is a significant advancement, but the historical literature should be referenced. **Geochronologists have been breaking and dating fragments of grains since the 1990's (e.g. Amelin, 1998;** *Chemical Geology*). **We have now included mention of these historical applications within the piece of text discussing limitations of traditional ID-TIMS.**

Line 87 – you indicate that laser cutting would induce localized elemental fractionation – would this be avoided by using a fempto second laser? **Laser cutting could be feasible for some grains, though would require additional testing on the fractionation effects of a fempto second laser that is beyond the scope of this study. The technique, however, if shown to drive minimal fractionation of U-Pb isotope systematics, would be a potentially useful tool for isolating larger crystals and grain domains, for example in zircon. We note, however, that the capability of the FIB to mill away complex, micrometer scale domains allows for confident removal of all host material.**

Line 125 – Heaman 2009, all of the ages in the Heaman 2009 study come from one single baddeleyite megacryst, are the grains measured in this current work from the same megacryst? It's not exactly clear from what you say. **The grain analysed here is not the same megacryst, though was extracted from the same sample as Heaman 2009.**

Line 127 – (58/68) – this needs correcting **Removed.**

Line 128 – I guess this is for the editors of the journal, but its likely that you don't need to explain the discordance calculation, if you want to do it, put the proper calculation. **Removed for simplicity.**

Line 135 – This is slightly confusing wording for the readers and it is also related to problems using discordance as a metric. You mention that 77% of the analyses by Ibanez-Mejia are >1% discordant, but are the errors on these analysis overlapping with the Concordia curve? **Reworded to "while the majority (77%) of U-Pb analyses are >1 % discordant outside of uncertainty (Ibanez-Mejia et al., 2014)".**

Line 187 – how did you measure mass fractionation for each cycle on the mass spectrometer if you didn't use a double spike? **This has been corrected in section 2.2 to indicate that a thermal mass fractionation correction of 0.1% per atomic mass unit for both Pb and U was made.**

Line 195 – I'm sure this doesn't make a difference to the end results – but why did you assume an average crystal Th/U of the magma since these crystals are not exactly from "an average" magma composition. **As the reviewer notes, it does not matter for the results. The Th/U assumption is a first-order correction that is routinely employed in our laboratory, and true, not always tailored to each sample dated (it will of course matter in much younger rock samples).**

Line 231 – you say that the effects of FIB extraction on U-Pb isotope systematics have never been explored – what about atom probe analysis? (the lead author of this work previously published a paper on baddeleyite U-Pb systematics on samples extracted by FIB techniques and then measured on the atom probe) **We address this comment on L255-259 with a new piece of text:** *"Previous studies utilising the extraction of baddeleyite domains using FIB instruments (such as for structural and isotopic analysis by atom probe tomography (APT); Reinhard et al., 2017; White et al., 2017a,b) provide a poor comparison, given the application of a low-energy (~40 pA, 5 kV) final polish to remove material that may have been damaged or implanted with Ga-ions during interaction with the beam."*

Lines 236-238 – it would be much better to show graphically the results here rather than explaining them. Ideally you would have a graph that shows age/discordance/common Pb or some other metrics vs the different methods used. Currently it is not entirely convincing that the extraction techniques do not induce some U-Pb disturbance. **We have now included the full data table within the manuscript (Table 1), which allows for a more complete comparison of age, discordance and common Pb measurements across all isotopic analyse.**

Section 4.2 – isotopic heterogeneity in Phalaborwa baddeleyite. It is known that the Phalaborwa baddeleyite grains contain some amount of Pb loss (and this may be endemic to baddeleyite in general – see Schaltegger and Davies 2018, Davis and Davis 2017). Therefore it seems a bit counter intuitive to use the U/Pb ratio of crystals to discuss isotopic heterogeneity. Its not clear if your discussion here is related to Pb loss or real age variation? I think this section needs to be reworded to make it more clear in that regard. Also, the Phalaborwa baddeleyite is a reference material not a standard (and you should refer to it as such throughout the paper). You also mention some degree of "care" (line 263) that should be taken when doing small volume U-Pb work on phalaborwa. What exactly do you mean by this? Do you have a recipe that should be followed to ensure that the best measurements can be made – this would be an interesting and useful addition to the paper. **Reworded to read more explicitly:** *"Care must be taken to select pristine subdomains of material when…..".* **Beyond this, the level of care applied to FIB-TIMS sample selection and preparation is identical to typical TIMS analysis.**

Lines 269-281 – You cite the work of Davis and Davis 2017, which discusses alpha recoil effects on baddeleyite ages. This current study could have been the perfect test case for the idea of Davis and Davis since you could have dated at high precision areas of baddeleyite crystals at the rim and at the centre of a large crystal. There are also some confusing sentences in this paragraph that should be corrected slightly. For example, the last sentence says – "allowing the targeted extraction of centralized regions which are unlikely to have lost Pb during an alpha recoil event" - the central regions of grains will have experienced Pb redistribution due to alpha recoil, but will not have lost Pb since adjacent areas will have ejected their Pb into the central region. I think you mean that there will have been no alpha recoil ejection from the centre of the grain. **Changed to ejected.**

Line 285 – add Rioux et al. 2010 (Contributions Mineralogy Petrology) here, you can also add this in the introduction since it's an important piece of work on baddeleyite U-Pb geochronology **Done.**

Lines 291 to 295 – the other reviewers raised concerns about the common Pb in some of the analysis here and their questions adequately covered my concerns. **We hope that our responses to the other reviewers address any and all concerns.**

Lines 296-299 – this is because you are doing TIMS analysis – I'm not sure that it is relevant to say that certain problems only associated with SIMS analysis are avoided by this technique – you are not avoiding these problems because these problems are not associated with your technique. **Orientation effects are a real problem for in-situ analysis of baddeleyite, as for SIMS you're stuck with the orientation of the grain. Highlighting that, by using FIB-TIMS, you can conduct in-situ analysis without the risk of orientation related effects is a strength of the technique.**

We hope these alterations and responses address any and all concerns with the manuscript.

Sincerely,

Lee Francis White (on behalf of all co-authors)

[revised manuscript text omitted]
$ (c) | 2σ | $^{206}Pb/^{238}U$ (c) | 2σ | error corr | $^{207}Pb/^{206}Pb$ (d) | 2σ | $^{206}Pb/^{238}U$ (d) | 2σ | $^{207}Pb/^{235}U$ (d) | 2σ | $^{207}Pb/^{206}Pb$ (d) | 2σ | % Disc (f) |
|---|---|---|---|---|---|---|---|---|---|---|---|---|---|---|---|---|---|---|---|---|
| | | | | | | | | | | | | | | | | Age (Ma) | | | | |
| 1 | Whole grain #1 | 0.60 | 431 | 0.02 | 0.17 | 36226 | 6.580 | 0.016 | 0.37480 | 0.00076 | 0.944 | 0.127333 | 0.000106 | 2052.0 | 3.6 | 2056.7 | 2.1 | 2061.5 | 1.5 | 0.5 |
| 2 | Whole grain #2 | 1.50 | 277 | 0.01 | 0.18 | 54821 | 6.574 | 0.015 | 0.37458 | 0.00071 | 0.940 | 0.127281 | 0.000105 | 2050.9 | 3.3 | 2055.8 | 2.0 | 2060.8 | 1.5 | 0.6 |
| 3 | 5 chips from mount | 0.80 | 1591 | 0.01 | 1.09 | 28030 | 6.581 | 0.017 | 0.37479 | 0.00086 | 0.908 | 0.127345 | 0.000137 | 2051.9 | 4.0 | 2056.8 | 2.3 | 2061.6 | 1.9 | 0.6 |
| 4 | 1 chip from mount | 1.30 | 3027 | 0.01 | 12.3 | 7104 | 6.066 | 0.015 | 0.34531 | 0.00072 | 0.920 | 0.127400 | 0.000132 | 1912.2 | 3.4 | 1985.3 | 2.2 | 2062.4 | 1.8 | 8.4 |
| 5 | 50x50x50 um cube (Ga⁺-FIB) | 0.40 | 443 | 0.02 | 1.91 | 2143 | 6.261 | 0.023 | 0.35705 | 0.00092 | 0.779 | 0.127179 | 0.000299 | 1968.2 | 4.4 | 2013.0 | 3.3 | 2059.3 | 4.2 | 5.1 |
| 6 | Flake (subset of rectangle #2) (Xe⁺-ρ FIB) | 0.20 | 106 | 0.01 | 0.33 | 1163 | 6.487 | 0.055 | 0.37174 | 0.00267 | 0.885 | 0.126561 | 0.000503 | 2037.6 | 12.5 | 2044.1 | 7.5 | 2050.7 | 7.0 | 0.7 |
| 7 | Flake (subset of rectangle #2) (Xe⁺-ρ FIB) | 0.05 | 254 | 0.01 | 0.67 | 464 | 6.413 | 0.066 | 0.36649 | 0.00356 | 0.906 | 0.126902 | 0.000557 | 2012.8 | 16.8 | 2034.0 | 9.1 | 2055.5 | 7.7 | 2.4 |
| 8 | 200x50x50 um rectangle #1 (Xe⁺-ρ FIB) | 2.50 | 284 | 0.02 | 0.66 | 24743 | 6.248 | 0.017 | 0.35613 | 0.00085 | 0.937 | 0.127252 | 0.000120 | 1963.8 | 4.0 | 2011.3 | 2.3 | 2060.4 | 1.7 | 5.4 |
| 9 | 100x100x100 um cube (Xe⁺-ρ FIB) | 5.00 | 397 | na | 0.54 | 83466 | 6.201 | 0.015 | 0.35327 | 0.00073 | 0.954 | 0.127302 | 0.000100 | 1950.2 | 3.5 | 2004.5 | 2.1 | 2061.1 | 1.4 | 6.2 |
| 10 | 200x50x50 um rectangle #2 (Xe⁺-ρ FIB) | 2.50 | 352 | 0.02 | 1.28 | 15456 | 6.161 | 0.015 | 0.35104 | 0.00073 | 0.908 | 0.127288 | 0.000129 | 1939.6 | 3.5 | 1998.9 | 2.1 | 2060.9 | 1.8 | 6.8 |
| 11 | 5x15um domain (Xe⁺-ρ FIB) | 0.08 | 510 | 0.04 | 0.25 | 3499 | 5.672 | 0.023 | 0.32430 | 0.00115 | 0.903 | 0.126843 | 0.000218 | 1810.7 | 5.6 | 1927.1 | 3.4 | 2054.7 | 3.0 | 13.6 |

NOTES:

[revised manuscript text omitted]

Section Break (Next Page)

| Page 8: [1] Formatted | Lee White | 23/04/2020 14:34:00 |

Font: (Default) Times New Roman

| Page 8: [2] Deleted | Sandra Kamo | 01/04/2020 09:39:00 |

| Page 8: [3] Deleted | Sandra Kamo | 03/04/2020 16:33:00 |

| Page 8: [4] Deleted | Sandra Kamo | 27/03/2020 16:03:00 |

| Page 8: [5] Deleted | Sandra Kamo | 01/04/2020 09:43:00 |

| Page 8: [6] Formatted | Lee White | 23/04/2020 14:34:00 |

Font: (Default) Times New Roman

| Page 8: [6] Formatted | Lee White | 23/04/2020 14:34:00 |

Font: (Default) Times New Roman

| Page 8: [7] Deleted | Sandra Kamo | 03/04/2020 16:47:00 |

| Page 8: [7] Deleted | Sandra Kamo | 03/04/2020 16:47:00 |

| Page 8: [7] Deleted | Sandra Kamo | 03/04/2020 16:47:00 |

| Page 8: [7] Deleted | Sandra Kamo | 03/04/2020 16:47:00 |

| Page 8: [8] Formatted | Lee White | 29/04/2020 12:39:00 |

Superscript

| Page 8: [8] Formatted | Lee White | 29/04/2020 12:39:00 |

Superscript

| Page 8: [8] Formatted | Lee White | 29/04/2020 12:39:00 |

Superscript